# Functional and Taxonomic Effects of Organic Amendments on the Restoration of Semiarid Quarry Soils

Isabel Miralles,[a] Raúl Ortega,[a] André M. Comeau[b]

[a]Department of Agronomy & Center for Intensive Mediterranean Agrosystems and Agri-food Biotechnology (CIAIMBITAL), University of Almeria, Almería, Spain
[b]Integrated Microbiome Resource (IMR), Dalhousie University, Halifax, Nova Scotia, Canada

Isabel Miralles and André M. Comeau contributed equally to this work.

**ABSTRACT** The application of organic amendments to mining soils has been shown to be a successful method of restoration, improving key physicochemical soil properties. However, there is a lack of a clear understanding of the soil bacterial community taxonomic and functional changes that are brought about by these treatments. We present further metagenomic sequencing (MGS) profiling of the effects of different restoration treatments applied to degraded, arid quarry soils in southern Spain which had previously been profiled only with 16S rRNA gene (16S) and physicochemical analyses. Both taxonomic and functional MGS profiles showed clear separation of organic treatment amendments from control samples, and although taxonomic differences were quite clear, functional redundancy was higher than expected and the majority of the latter signal came from the aggregation of minor (<0.1%) community differences. Significant taxonomic differences were seen with the presumably less-biased MGS—for example, the phylum *Actinobacteria* and the two genera *Chloracidobacterium* (*Acidobacteria*) and *Paenibacillus* (*Firmicutes*) were determined to be major players by the MGS and this was consistent with their potential functional roles. The former phylum was much less present, and the latter two genera were either minor components or not detected in the 16S data. Mapping of reads to MetaCyc/BioCyc categories showed overall slightly higher biosynthesis and degradation capabilities in all treatments versus control soils, with sewage amendments showing highest values and vegetable-based amendments being at intermediate levels, matching higher nutrient levels, respiration rates, enzyme activities, and bacterial biomass previously observed in the treated soils.

**IMPORTANCE** The restoration of soils impacted by human activities poses specific challenges regarding the reestablishment of functional microbial communities which will further support the reintroduction of plant species. Organic fertilizers, originating from either treated sewage or vegetable wastes, have shown promise in restoration experiments; however, we still do not have a clear understanding of the functional and taxonomic changes that occur during these treatments. We used metagenomics to profile restoration treatments applied to degraded, arid quarry soils in southern Spain. We found that the assortments of individual functions and taxa within each soil could clearly identify treatments, while at the same time they demonstrated high functional redundancy. Functions grouped into higher pathways tended to match physicochemical measurements made on the same soils. In contrast, significant taxonomic differences were seen when the treatments were previously studied with a single marker gene, highlighting the advantage of metagenomic analysis for complex soil communities.

**KEYWORDS** shotgun metagenomics, functional profiles, bacterial taxonomy, soil restoration, sewage sludge, compost, semiarid mining soils

Address correspondence to Isabel Miralles, imiralles@ual.es.

mSystems®

**S**oils represent one of the main sources of microbial diversity, containing between 2,000 and 8.3 million bacterial species per gram of soil (1, 2). Their taxonomic diversity is also mirrored by the diversity of their protein-encoded functions (3)—microbial communities play key roles in maintaining multiple ecosystem functions and services such as the cycling of nutrients (for example, carbon, nitrogen, etc.), participating in primary production, and regulating soil fertility and plant health (3–8). They participate in, or are solely responsible for, degradation, transformation, and biosynthesis reactions in biogeochemical cycles, as well as the detoxification of natural and human-made pollutants (9).

Human impacts that degrade physical and chemical soil qualities also affect soil microbial communities, reducing their overall diversity (10–16). These impacts are especially drastic in soils affected by extractive activities in quarries (14, 16), where superficial organic and/or organo-mineral horizons, which generally contain the largest number of soil microorganisms (17, 18), are eliminated from the soil (16, 19). Such disturbances could influence the correct functioning of soil biogeochemical cycles in the underlying soils. Nevertheless, the disappearance of some microbial species due to soil degradation may not have the same impact on basic or widely distributed functions across living organisms, such as global organic matter decomposition and N and P cycling (20–22), because these same functions can be performed by many different representatives. This phenomenon, known as functional redundancy, contributes to the maintenance of the stability and functioning of the whole soil ecosystem, which has a high buffering capability, and is sometimes referred to as "soil memory" (20). However, the loss of specialized or rare functions carried out by smaller groups of specialized species, such as those performing detoxification, methanogenesis, or mineralization of recalcitrant organic compounds (9, 23), could lead to the collapse of different metabolic pathways, resulting in a strong negative impact on overall ecosystem functioning and services (20, 23).

Soil microbial communities, in addition to being clearly affected by processes such as mining practices in arid environments (14, 16, 24), also suffer significant impacts from subsequent restoration treatments necessary for the recovery of soil functionality (16, 24–26). The application of organic amendments to mining soils has been shown to be a successful method of restoration, improving physical (e.g., moisture and aggregate stability) and chemical (e.g., pH, organic matter, total N, available P, etc.) soil properties and key indicator enzymatic activities (involved in the C, N, and P cycles) (19, 24, 26–29) and influencing $CO_2$ fluxes (30). After the application of organic amendments in mining soils, the changes in the abovementioned physicochemical soil properties (which are considered key factors in directing the structure of soil microbial communities) (31–36) indirectly influence soil bacterial community composition (16, 24). Moreover, the organic amendments themselves can directly affect the composition of the bacterial communities of the soils through the inoculation of new bacteria present in the amendments or due to changes in the chemical composition of the amendments, with a higher content of labile or recalcitrant organic compounds allowing the proliferation of some bacterial communities over others (16, 24). These changes to the (taxonomic) composition of microbial communities will foreseeably affect the functions that these microorganisms perform in the soils. For example, the application of compost organic matter amendments favors specialized functions such as lignin degradation (23), requiring the cooperation of a diverse group of microorganisms (37).

Despite the large diversity of microbial communities in the enormously heterogeneous matrices of soils, their crucial roles in soil biogeochemical cycles, and their complex taxonomy-function relationships, thorough metagenomic studies in severely human-impacted soils are less common. In general, soil microbial diversity has been mainly studied using 16S rRNA gene analyses (here referred to as 16S) until most recently (38–40), but this method provides only taxonomic information. However, in the last decade, the application of culture-independent (meta)genomics approaches (37, 41) based on high-throughput DNA sequencing has proved a promising tool to investigate the

abundance of specific genes, entire functional profiles, and ecological significances of microbial communities living in diverse ecosystems (3, 42–51). As metagenomic analysis forgoes using selective primers and PCR, it is widely considered a much less biased approach to analyze communities—although challenges still remain in having adequate database coverage of novel organisms, the preponderance of unknown functions, and the fact that gene presence does not necessarily equate to expressed function (52, 53). That being said, given the greater resolution of this technique, the main goal of the work presented here is the further functional profiling of the effects of different soil restoration treatments (of differing organic amendments from recycled materials) applied to degraded, arid quarry soils in southern Spain which had previously only been molecularly profiled with 16S analyses (16) but which have been extensively studied from a physicochemical/biological activity perspective as part of a multiannual restoration experiment (26, 30). The organic amendments applied were (i) 100% vegetable compost manufactured from garden waste (COVG), (ii) 100% vegetable compost manufactured from horticultural greenhouse crop waste (COHort); (iii) treated sewage sludge waste (SS), (iv) an amendment made from a 50:50 mixture of COVG + SS, and (v) an amendment made from a 50:50 mixture of COHort + SS. Changes in the taxonomic composition and functions, especially related to the carbon cycle, were examined in detail between soils restored with the different treatments compared to degraded unrestored soils in the quarry (Control) and surrounding natural soils undisturbed by mining activity (Natural). Given the results of the previous studies on these restoration experimental plots, our working hypotheses were (i) that experimental and control plots within the degraded soils would show stark differences from nearby natural soils; (ii) that the same major players detected in the 16S analyses would also be recovered in the metagenomic taxonomic analysis; (iii) that the functional data unique to metagenomics would discriminate among soil types, while still displaying a minor amount of functional redundancy between them; and (iv) that the SS treatment would show the highest functional diversity due to its previously observed highest biological activity of all the treatments (26, 30).

## RESULTS

**Soil taxonomic composition.** The overall pattern of taxonomic distributions between soils showed very clear separation of each primary treatment (COHort, COVG, or SS) from the control and natural soils in ordination (Fig. 1), and the ADONIS results confirmed that 87% of the variation could be attributed to the treatment category. The replicates for the control and natural soils were very tightly clustered, and these two soil types were also relatively close to each other. There was slightly more spread among the treatment replicates (widest for the SS); however, they were also distinctly clustered apart so that no primary treatments overlapped. The two treatment mixtures (COHort + SS and COVG + SS) displayed "intermediate" patterns where their centroid distances were being pulled more toward the mixed-in SS position; however, the effect of adding SS to COVG appears to have had a much greater impact, as COHort + SS shows a minor shift at best. These clustering patterns were driven by over 200 significantly different genera between treatments and untreated plots (Naturals + Controls), whether in the ALDEx2 or linear discriminant analysis (LDA) effect size (LEfSe) analysis (Fig. 2). However, a substantial portion of those differences were due to minor genera (i.e., rare/low-abundance genera), and when individual treatments were examined on their own, a range of ∼20 to 70 genera were found to be enriched in each. In the ALDEx2 comparisons, the SS amendment showed the most substantial differences from the control soils, with COHort showing intermediate differences and COVG being the closest to control soils. Perhaps not surprisingly given their near-overlap in ordination, the natural soils showed zero significantly different genera compared to the study control plots, and the two plot types were then combined into the "untreated" plots for the majority of the taxonomic and functional comparisons below. As the mixture plots (COHort + SS and COVG + SS) simply showed straightforward intermediate patterns, these two plot types were excluded from some of the subsequent

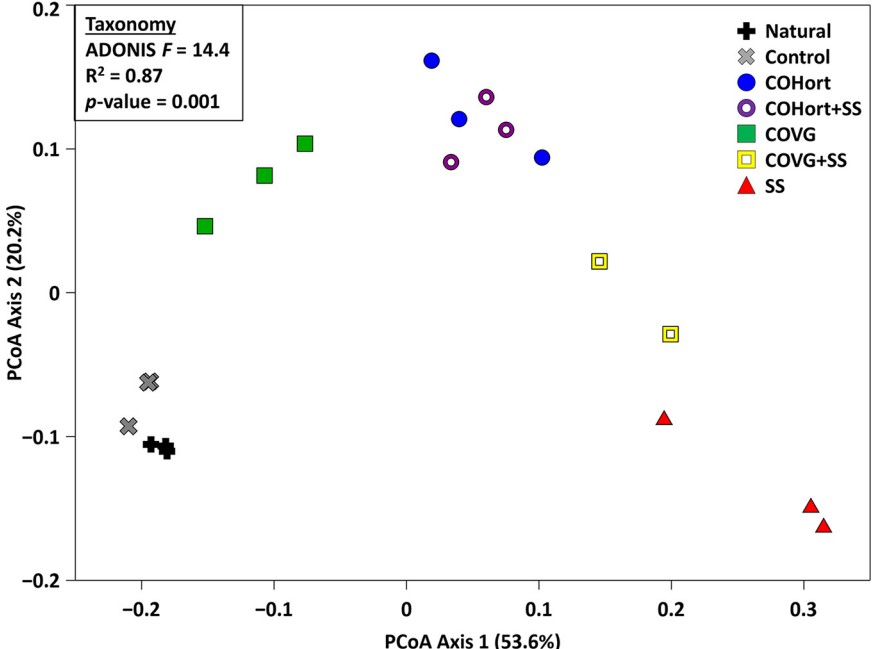

**FIG 1** Overall patterns of restoration treatment and untreated plots derived from the taxonomic abundances. Bray-Curtis PCoA of the Kraken2 (Bracken-corrected) individual sample counts of genera found ($n$ = 329 across all samples after low-abundance filtering), color coded (along with specific symbols) by treatments or controls with triplicates for each soil plot type. Results of the ADONIS test for significant groupings are included in the figure inset. COHort, plots treated with vegetable compost from horticultural waste; Control, plots of quarry soils without restoration amendments; COVG, plots treated with vegetable compost from garden waste; Natural, non-human-impacted natural soils near the study area; SS, plots treated with treated sewage sludge; + SS, 50:50 mixtures of the preceding vegetable composts with SS.

analyses/plots (e.g., ALDEx2 and MetaCyc) in order to specifically zero in on the factors specific to the primary single treatments (COHort, COVG, and SS) compared to the controls. Although trending toward differences in diversity (Shannon's $H$) in the multiple comparison (Kruskal-Wallis), there were ultimately no significant differences between any of the study plots in pairwise comparisons due to multiple test correction (Fig. 2).

In term of high-level taxonomy, the distributions of the phyla in the soils (see Fig. S1 in the supplemental material) also showed fairly clear patterns of untreated soils being distinct from the plant-based amendments (COHort and COVG), which were also themselves quite different from the sewage amendment. *Actinobacteria* and *Proteobacteria* were the two largest contributors to all communities (>50% combined in all soils), although skewed toward the former in untreated/COHort/COVG and toward the latter in SS. *Bacteroidetes* and *Firmicutes* were more prevalent in SS (~12 to 19%), followed by COHort and COVG (~8 to 12%), but were minor components in untreated soils (<3%). Conversely, *Acidobacteria* represented ~4% of the total community in untreated soils but were ~1% or less in all amendments.

At the more detailed genus level, the overall distributions of the most prevalent genera also reinforced the above partitioning of untreated versus COHort/COVG versus SS (Fig. S2). Differential analysis of these genera (Fig. 3 and 4) showed between 110 and 140 significant genera separating the untreated soils from the treatments, with almost complete agreement between ALDEx2 and LEfSe results; however, most of these were with small LDA scores or small effect sizes (<1% absolute difference in mean proportions). That being said, Fig. 3 and 4 show those genera with the most significant effects and abundances across the various plots. *Streptomyces* was the most abundant single genus across all soils and was the top significantly enriched genus in untreated soils, followed as a close second (both >4.0 log LDA) by *Chloracidobacterium* (*Acidobacteria*),

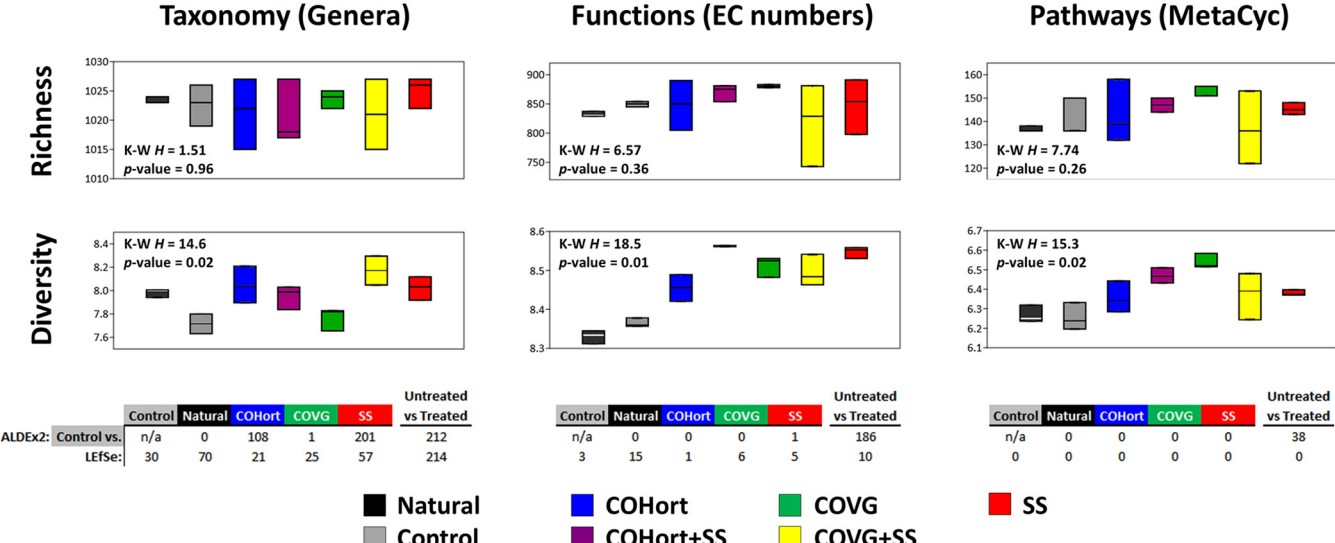

**FIG 2** Overall richness, diversity, and differentially abundant features of restoration treatment and untreated plots derived from the taxonomic (left column), function (middle column), and pathway (right column) abundances. Top panels are the richness boxplots (showing median [bar] and range of triplicates) for each data type, with total numbers of different genera, EC numbers, or MetaCyc pathways detected per sample represented on the $y$ axes. Middle panels are the Shannon's $H$ boxplots (as above) for each data type. For all boxplot panels, results of the Kruskal-Wallis tests for differences between medians are shown in the upper or lower left. Bottom tables are the numbers of differentially abundant/enriched features for each data type determined using ALDEx2 and LEfSe. Shown are the results of individual testing for each unmixed treatment plot or, in the last column of the tables, for the differences between untreated plots (Controls + Naturals) and all five treated plots together. Note that ALDEx2 requires pairwise comparisons; therefore, the feature numbers in the first five columns of each table are for the plots compared to the Controls, whereas LEfSe analyzes all plots at once to find the nonredundant features enriched in each plot. Color coding and treatment abbreviations are as in Fig. 1.

which averaged a 3% greater mean proportion in untreated soils (Fig. S2). The SS treatment showed the longest list of differential genera, dominated mostly by *Gammaproteobacteria* and *Bacteroidetes* versus *Alphaproteobacteria* and *Actinobacteria* for untreated soils, among which were *Lysobacter*, *Pseudomonas*, *Sphingobacterium*, *Pedobacter*, and *Flavobacterium*. The top abundant genera that were significantly enriched in the COHort treatments were *Paenibacillus* (*Firmicutes*; very abundant in both plant amendments [Fig. 4]), *Isoptericola* (*Actinobacteria*), and *Nocardiopsis* (*Actinobacteria*), among a few more from diverse phyla. COVG treatments showed further enrichment of various *Actinobacteria*, including *Nocardioides*, *Mycobacterium*, *Mycolicibacterium*, *Microbacterium*, and *Promicromonospora*.

**Soil functional profiles.** Similar to the overall taxonomic patterns, the ordination plots of the distributions of both the individual functions (from EC numbers; Fig. 5A) and those functions grouped into their respective pathways (Fig. 5B) showed the same distinct clustering of the untreated soils away from the amendments (although some replicates were closer now), and the individual primary treatment types (COHort, COVG, and SS) still separated from each other (accounting for 67 to 71% of the total variation by ADONIS). The mixed treatments (+ SS) showed more overlap functionally here, but the sewage treatment still showed the greatest distance from controls. ALDEx2 pairwise comparisons resulted in many fewer significantly different features compared to taxonomy, falling to almost none in treatments versus the control soils, and LEfSe enriched features were absolutely 0 for pathways and a few functions for each treatment (Fig. 2). Additionally, unlike the top taxonomy features, most of these functional differences were small—LDA scores were mostly near the minimum of 2.0 (Fig. 6) and ALDEx2 effect sizes were also almost all <2.0. Similar to taxonomy, the overall functional diversity trended toward higher values in treatments, but no significant differences were found in the pairwise comparisons after multiple test correction (Fig. 2).

When each treatment was compared to the untreated controls, some small functional differences were seen, with 1 to 6 enriched EC numbers in controls and plant-based amendments and only 5 differences for the sewage treatment (Fig. 6). The

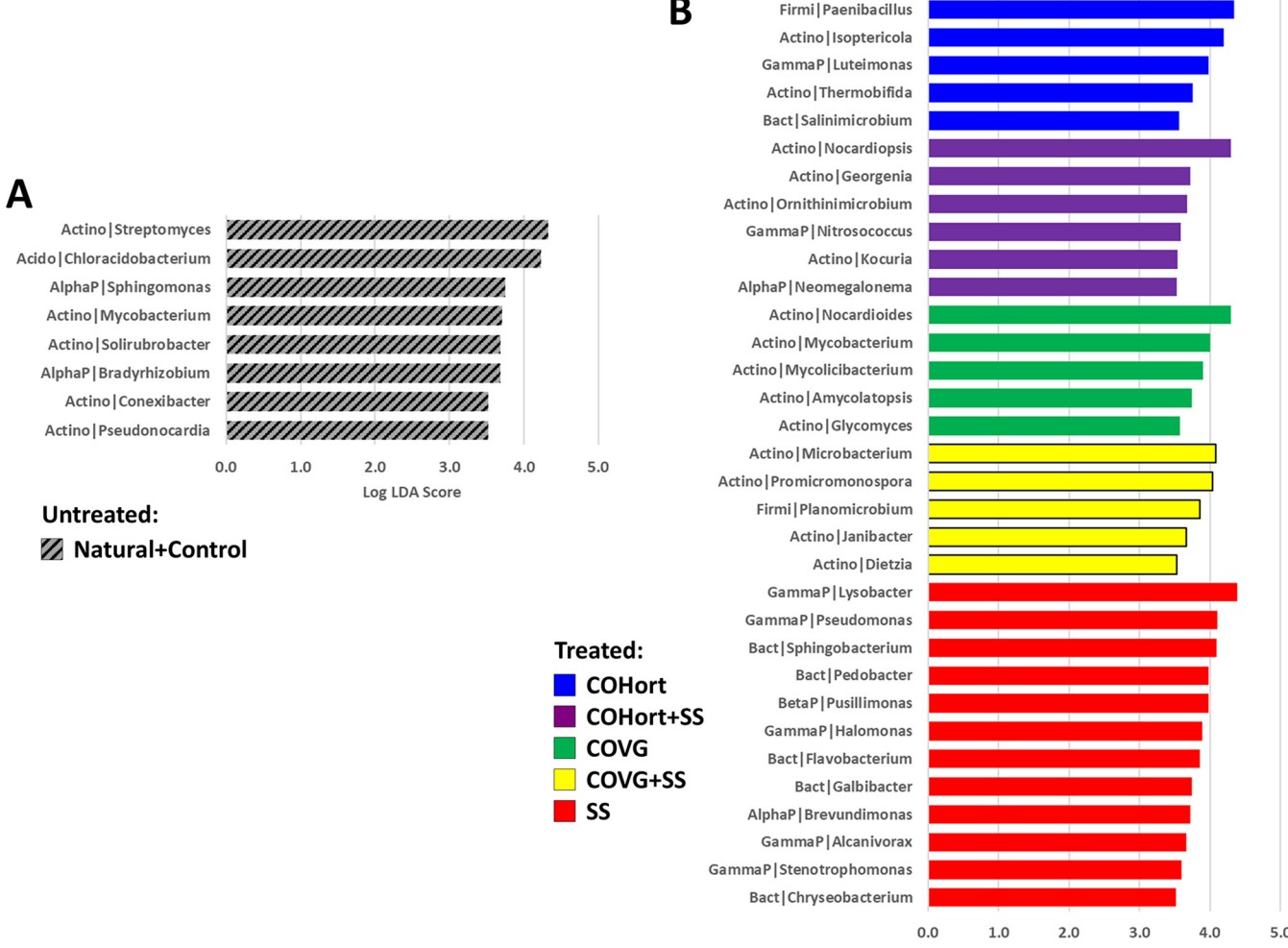

**FIG 3** Top significantly enriched genera in restoration treatment and untreated plots. Due to the large numbers of significant features found for each plot type (tables in Fig. 2), only the top LEfSe-determined enriched genera with log LDA scores of >3.5 are shown here for the untreated plots (Controls + Naturals) (A) and the treated plots (B), further broken down by treatment type using the same color coding and treatment abbreviations as in Fig. 1. Note that most of these genera were also found to be differentially abundant in the ALDEx2 comparisons (Fig. 4 gives a comprehensive comparison). Acido, *Acidobacteria*; Actino, *Actinobacteria*; AlphaP, *Alphaproteobacteria*; Bact, *Bacteroidetes*; BetaP, *Betaproteobacteria*; Firmi, *Firmicutes*; GammaP, *Gammaproteobacteria*.

untreated soils showed slightly higher abundances of four functions related to various protein/amino acid functions, such as glutaminase and endopeptidase, as well as urease degradation and phosphate transport. COHort showed the single enrichment of ectoine synthase, whereas COVG showed enrichment of 6 functions, including proteosome functions (e.g., Pup ligase) and energy/metabolism functions, such as glutamate-ammonia ligase from the N cycle and some cytochrome + NADH functions (mostly in the + SS mixture). The sewage treatment was enriched in 5 functions, among which were thymidine and lipid A synthesis (EC 3.5.1.108), rRNA/ribosome maturation (EC 2.1.1.166), and sodium motive force for flagella or transport (EC 7.2.1.1).

In terms of pathways, ALDEx2 found 6 significantly more abundant in the treated soils and 32 more abundant in untreated plots (Fig. 7), although effect sizes were quite small and LEfSe did not concur, having found 0 functional differences. Treated soils showed more pyridoxal 5′-P, pyrimidine, and lipid biosynthesis, as well as slightly more abundant denitrification pathways. The small shifts in untreated soil pathways were mostly in various isoleucine and arginine biosynthesis pathways; various biosynthesis pathways involving peptidoglycans, salvage pathways, and queuosine biosynthesis; and energy pathways such as glycolysis, tricarboxylic acid (TCA) cycles, and a sulfate assimilation pathway.

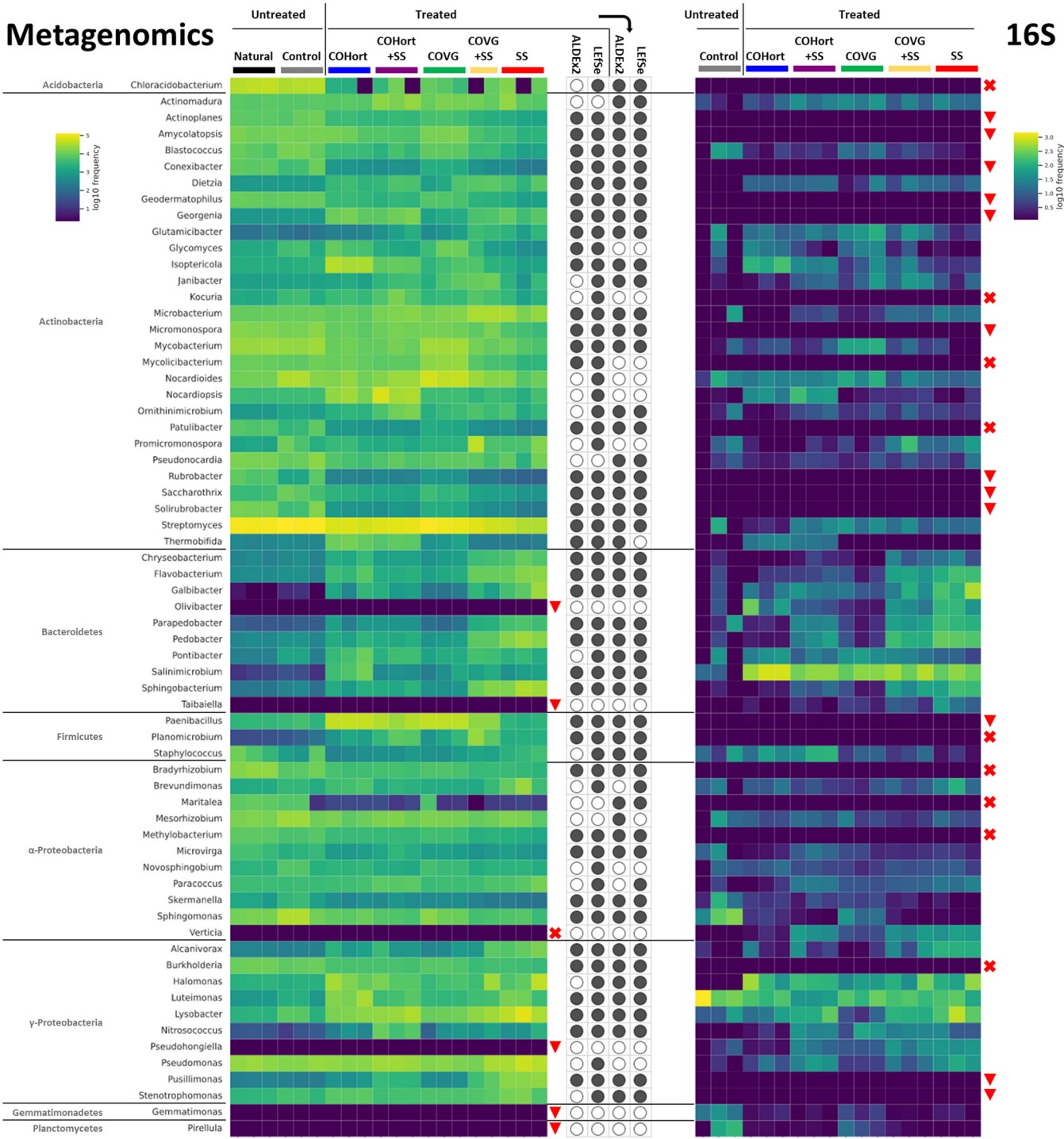

**FIG 4** Top genera in restoration treatment and untreated plots from metagenomic sequencing (left heatmap) and previous 16S rRNA gene sequencing (right heatmap) of the same samples (blue shows less and yellow shows more for both heatmaps). Also indicated are whether the metagenomic genera were found to be differentially abundant/enriched by either ALDEx2 or LEfSe (center: black-filled dots for yes, empty for no), either in individual testing for each plot (first and second dots) or for the differences between untreated plots versus all five treated plots together (third and fourth dots), as in Fig. 2. To be included in these heatmaps, genera had to be either (i) among the top 5 metagenomic LEfSe-enriched genera for each treatment or untreated plot or (ii) among the top 20 metagenomic LEfSe-enriched genera for untreated or treatment plots combined or (iii) all of the 25 genera (*Aminobacter* was removed since it was <1% abundant across all 16S samples) identified by the network analysis of Rodríguez-Berbel et al. (16) as having importance in structuring the 16S data (only identified genera; unclassified taxa removed). If a genus was not detected in one or the other data set, a red X is marked to the right of the respective heatmap. Downward red triangles indicate that those genera were at <1% abundance (summed across all samples) for that data set and were therefore excluded from the respective heatmap. Sample color coding and treatment abbreviations are as in Fig. 1.

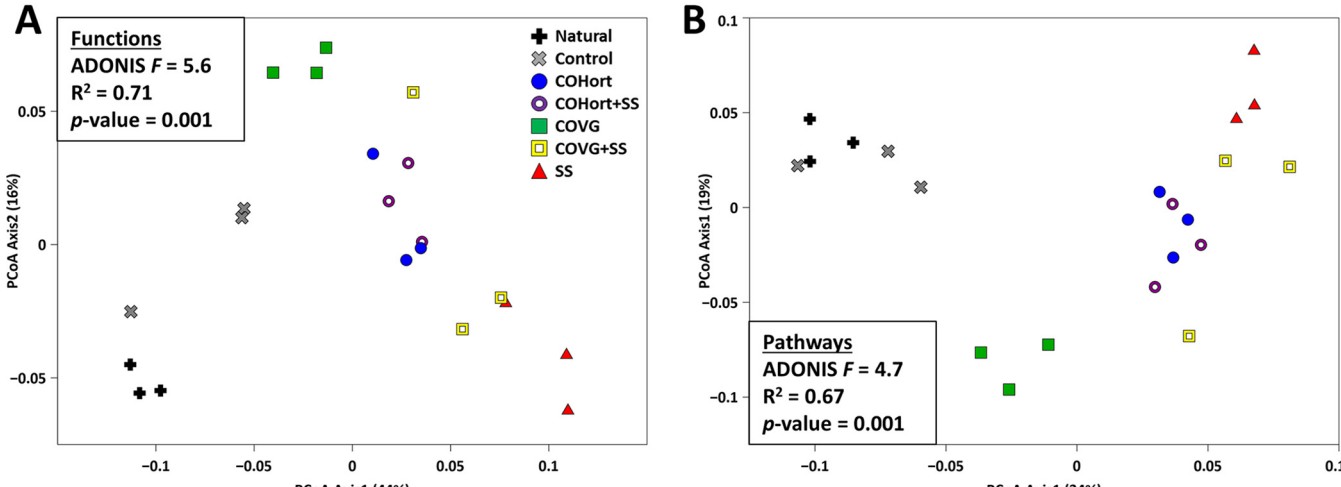

**FIG 5** Overall patterns of restoration treatment and untreated plots derived from the functional abundances. (A) Bray-Curtis PCoA of the individual sample counts of all EC functions found ($n = 376$ across all samples after low-abundance filtering). (B) Bray-Curtis PCoA of the individual sample counts of all MetaCyc pathways found ($n = 148$ across all samples after low-abundance filtering). Results of the ADONIS tests for significant groupings are included in the figure insets. Color coding, symbols, and treatment abbreviations are as in Fig. 1.

Finally, when the various individual functions and pathways were grouped into the MetaCyc/BioCyc categories (Fig. S3 to S5), a general trend was observed of overall higher biosynthesis and degradation capabilities in all treatments versus control soils, with sewage showing the highest mapping of reads to those functions and COHort/COVG being at intermediate levels. For example, this order was mostly maintained through all major categories of biosynthesis, such as amino acid and lipid synthesis (Fig. S3A). Interestingly, for carbohydrate biosynthesis, the order was the same for most subcategories, with a large spike in sugar-nucleoside synthesis for SS, but essentially inversed for glycan and glycogen synthesis (Fig. S3B). Mostly all of the degradative categories followed the SS > (COVG/COHort) > control pattern (Fig. S4), including the more abundant amino acid and carbohydrate degradation subcategories, as well as the more minor aromatic degradation subcategory. In the energy and "other pathways" categories, substantial numbers of reads mapped to fermentation, $CO_2$ fixation, and $C_1$ compound utilization (Fig. S5), again higher in treatments than control soils. In terms of inorganic nutrient metabolism, only the N cycle showed substantial read counts, with control and SS at similar levels and only a small extra amount of mapping in the two plant-based amendments. Overall, though, it must be kept in mind that many of these cumulative count differences represent only about 10% variance between many of the treatments/controls.

## DISCUSSION

The overarching goal of this study was to further compare previously studied treatment effects on degraded quarry soils, but using more fine-scale, and supposedly less-biased, metagenomics. Comparisons with the 16S analysis are presented below, along with the newly determined functional data which allowed for better comparison with measured physicochemical processes and clearer understanding of the potential mechanisms at play.

**Similar microbial profiles between natural and control soils.** Before delving into the details of the compost treatment effects, it is appropriate to address our first hypothesis which also anticipated that the unamended control plots within the mining area would show substantial differences from the nearby natural soils that were undisturbed by mining activities. However, control and natural soils were nearly overlapping in ordination and showed no significant taxonomic or functional differences between them, leading us to group them together as "untreated" soils (to increase statistical robustness in the analyses). One potential explanation of this similarity is that the "soil

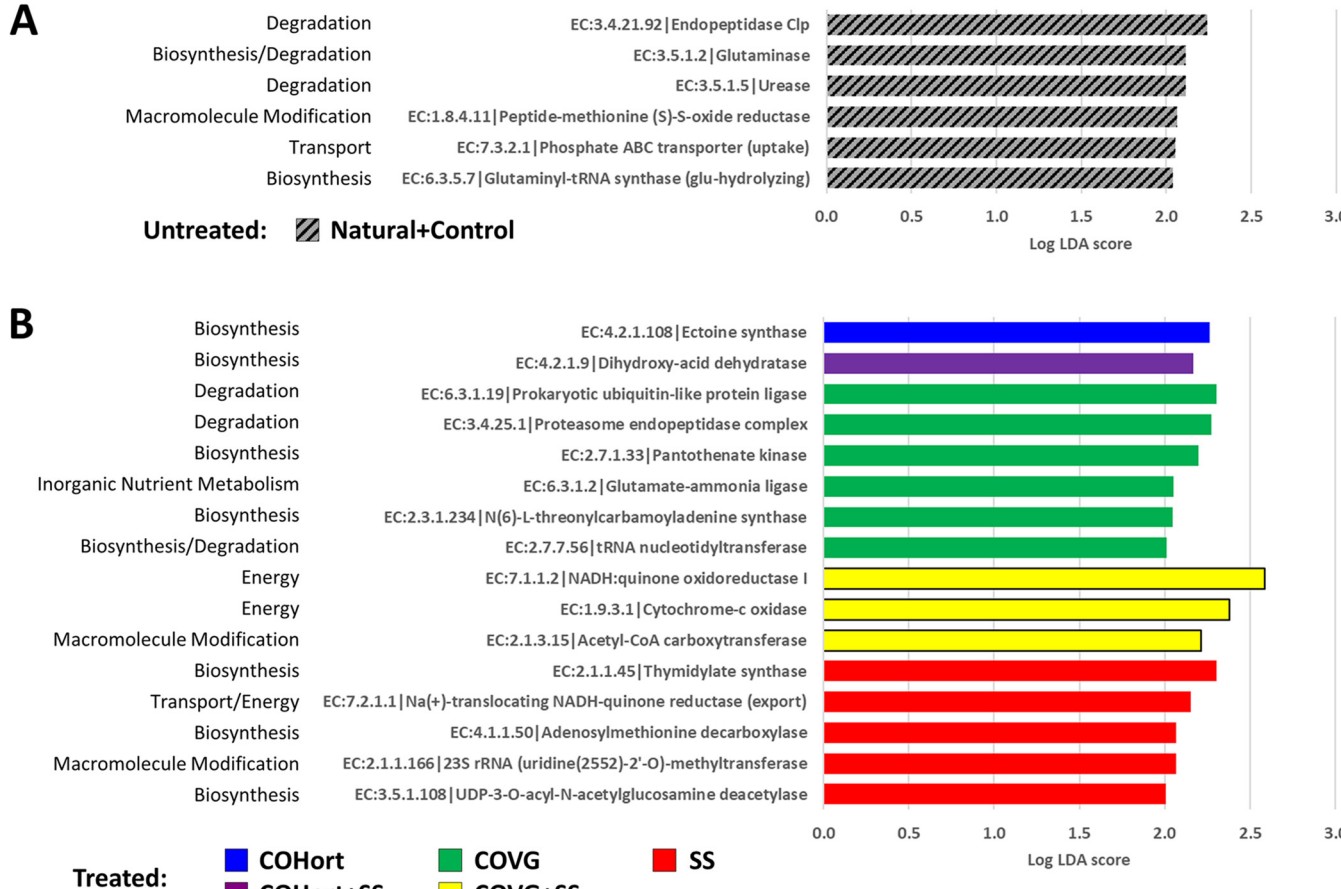

**FIG 6** Significantly enriched EC functions in restoration treatment and untreated plots. All LEfSe-determined enriched functions (default minimum log LDA 2.0) are shown here for the untreated plots (Controls + Naturals) (A) and the treated plots (B), further broken down by treatment type using the same color coding and treatment abbreviations as in Fig. 1. Functions are labeled on the far left with their corresponding MetaCyc/BioCyc broad categories (similar to Fig. S4 to S6).

memory"—which includes the collection of similar species capable of similar functions and dormant microbes (20, 54, 55)—in the lower soil horizons could rapidly reseed the mining soils after the end of active exploitation, so that their profiles now closely resembled nearby natural soils. A second potential explanation is that the very high dispersal rate of soil bacteria (54, 55) led to rapid recolonization of the mining soils with typical topsoil bacteria from the local geographic area. These two phenomena are not mutually exclusive, and both could be influencing primary soil succession/recovery happening in this zone since the cessation of mining activities. Additionally, as these soils are semiarid and have a shallow topsoil layer (a few centimeters) composed primarily of biocrusts and shrubs/grasses, they may require less time to return to a taxonomic profile resembling natural soils after perturbations compared to more organically complex temperate soils.

That being said, even though their microbial profiles appear similar, the degraded mining control soil plots have not exhibited spontaneous revegetation, and extraction of DNA was quite difficult compared to the natural soils, which concurred with previous measurements of bacterial biomass. The similarity to natural soils was therefore quite surprising given that every single measure of soil function (chemistry, $CO_2$ flux/respiration, enzyme activities, lipid/carbohydrate contents, etc.) indicated the control soils were of poor biological quality and only the compost treatment plots showed improvement for supporting introduced vegetation (16, 26, 30). It could therefore be that the only two things separating the control mining soils from assuming more "natural soil function" are the topsoil nutrients removed during mining (reintroduced in the compost plots) and the

## A — Pathways Enriched in Untreated Soils (Naturals+Controls; 32)

**Amino Acid Biosynthesis**
ARGSYNBSUB-PWY = L-arginine biosynthesis II (acetyl cycle)
ARGSYN-PWY = L-arginine biosynthesis I (via L-ornithine)
ARO-PWY = chorismate biosynthesis I
BRANCHED-CHAIN-AA-SYN-PWY
COMPLETE-ARO-PWY = aromatic amino acid synthesis superpathway
GLUTORN-PWY = L-ornithine biosynthesis I
ILEUSYN-PWY = L-isoleucine biosynthesis I (from threonine)
MET-SAM-PWY = superpathway of S-adenosyl-L-methionine biosyn.
PWY-3001 = superpathway of L-isoleucine biosynthesis I
PWY-5097 = L-lysine biosynthesis VI
PWY-5101 = L-isoleucine biosynthesis II
PWY-5103 = L-isoleucine biosynthesis III
PWY-5505 = L-glutamate and L-glutamine biosynthesis
PWY-7400 = L-arginine biosynthesis IV (archaea)
VALSYN-PWY = L-valine biosynthesis

**Degradation**
HISDEG-PWY = L-histidine degradation I

**Other Biosynthesis**
GLUCONEO-PWY = gluconeogenesis I
PANTO-PWY = phosphopantothenate biosynthesis I
PEPTIDOGLYCANSYN-PWY = PG monomer biosynthesis I
PWY-5188 = tetrapyrrole biosynthesis I (from glutamate)
PWY-6387 = UDP-*N*-NAM-pentapeptide biosynthesis I (PG subunit)
PWY-6545 = pyrimidine deoxyribonucleotides *de novo* biosynthesis III
PWY-6700 = queuosine biosynthesis I (*de novo*)
PWY-7094 = fatty acid salvage
PWY-7208 = superpathway of pyrimidine nucleobases salvage

**Energy**
GLYCOLYSIS
P42-PWY = incomplete reductive TCA cycle
PWY-101 = photosynthesis light reactions
PWY-5484 = glycolysis II (from fructose 6-phosphate)
PWY-5913 = partial TCA cycle (obligate autotrophs)
PWY-6969 = TCA cycle V (2-oxoglutarate synthase)
SO4ASSIM-PWY = assimilatory sulfate reduction I

## B — Pathways Enriched in Treated Soils (6)

PYRIDOXSYN-PWY = pyridoxal 5'-phosphate biosynthesis I
PWY-6151 = S-adenosyl-L-methionine salvage I
NAGLIPASYN-PWY = lipid IV biosynthesis
PWY-7187 = pyrimidine deoxyribonucleotides de novo biosynthesis II
DENITRIFICATION-PWY = nitrate reduction I
PWY-6748 = nitrate reduction VII (denitrification)

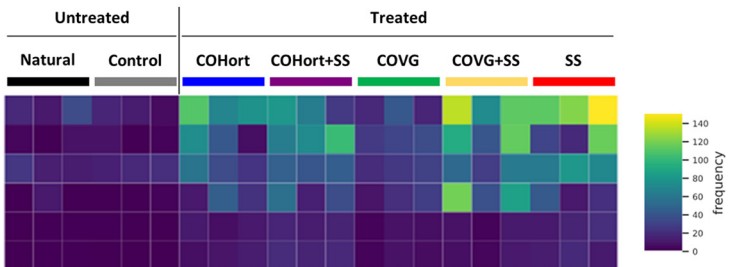

**FIG 7** Significantly enriched MetaCyc pathways in restoration treatment and untreated plots. ALDEx2-determined differential pathways are listed here for the untreated plots (Controls + Naturals) (A), grouped by their corresponding MetaCyc/BioCyc broad categories (similar to Fig. S4 to S6), and the treated plots (B), visualized by treatment type in the read count heatmap. Color coding and treatment abbreviations are as in Fig. 1. NAM, *N*-acetylmuramoyl; PG, peptidoglycan.

rates of growth plus expression of the same bacteria that are present in both, which could be elucidated by metatranscriptomics, as discussed in context below.

**Treatment-induced taxonomic shifts and comparison to 16S data.** The PCR-targeted 16S analysis of the exact same soil sample DNAs (16) and the assumedly less-biased shotgun metagenomic sequencing (MGS) analysis presented here do show significant overlap in community characterization (Fig. 4). However, there are also multiple differences in fine details that will be outlined below. In the first case, both methods showed distinct separation of each treatment type, based on overall taxonomic distributions, indicating that the amendments were responsible for significant community shifts from control soils. Whereas the 16S data found higher overall community diversity in COVG (and treatment mixtures) and lower diversity in control soils, no significant differences were found using the current MGS method. This discrepancy is perhaps a side effect of the difference in resolution of the two techniques: the MGS method tends to detect vastly more features than 16S, which then decreases the maximal variance that can be observed between categories in compositional data and also vastly increases the statistical impact of the multiple test correction burden by needing greater differences to pass the false-positive detection limits (56). Case in point, the 16S method detected 12 phyla and 152 genera among all soils, whereas the MGS method detected 49 phyla and 1597 genera at the same minimum abundance levels (>0.1% in all samples). Although one of the advantages of the untargeted MGS is the ability to also capture archaeal and eukaryotic information, and phyla from those two other domains were observed, the vast majority of the taxa from MGS were still bacterial (see Fig. S1 and S2 in the supplemental material); hence, the data set is still directly comparable to the 16S data. As a point of reference, the average ~1% Ascomycota (most abundant fungi) and

~0.5% *Euryarchaeota* (most abundant *Archaea*) across all samples are typical of their contribution levels in soils (57–59).

The top 5 most abundant bacterial phyla in all treatments for the 16S were (in order) *Bacteroidetes*, *Proteobacteria*, *Actinobacteria*, *Planctomycetes*, and *Firmicutes*, albeit with different proportions in each treatment—for example, *Proteobacteria* were dominant in controls versus *Bacteroidetes* in amended soils (especially SS) (16). Conversely, the order of the MGS top bacterial phyla was *Actinobacteria*, *Proteobacteria*, *Bacteroidetes*, *Firmicutes*, and *Acidobacteria*, with *Actinobacteria* dominant in all plots, except the sewage treatment, where *Proteobacteria* were most abundant. Although *Bacteroidetes* can be a dominant phylum in soils, it is typically less abundant and similar to the minor phylum *Planctomycetes* (60). It is not uncommon for 16S and MGS analyses in soil to provide differing taxonomic assessments (61); however, *Actinobacteria* have consistently been shown to be dominant in soils using MGS—examples include disparate locations such as the Florida Everglades and croplands in China (62, 63)—but even more pronounced in cold or dry desert soils (3), which are the soil type most comparable to the arid Spanish quarry soils in this study. If the MGS pattern is taken as more accurate, the overrepresentation of *Bacteroidetes* in 16S may be related to more cultured representatives in the reference databases having more strongly influenced the design of 16S primers or simply to greater inherent priming efficiency of this group. *Actinobacteria* may present lower 16S priming efficiencies; however, the inclusion of enough of their genomes in MGS databases seems to adequately identify their community contributions in this non-PCR-based method. Additionally, analyses of actinobacterial phylotypes in soil seem to indicate that a large proportion of them belong to "uncultured/unclassified" groups (64, 65), further indicating that current primers may not be capturing realistic actinobacterial contributions with 16S. The two methods do agree on the larger contributions of copiotrophic *Bacteroidetes* in SS and *Firmicutes* in COVG (and COHort for MGS); however, the greater contribution of copiotrophic *Proteobacteria* in SS shown by MGS is more consistent with their substantial dominance of sewage sludges (66, 67), upon which this amendment is based.

The presence of *Chloracidobacterium* (*Acidobacteria*) in the MGS data as a main marker for control soils is probably indicative of the nutrient-poor conditions of these nonamended soils. The 16S data missed the genus completely, indicating potential primer biases against the phylum as a whole. Members of the *Acidobacteria* are recalcitrant to cultivation; hence, many of their functions are unknown, due to their oligotrophic nature (68)—accordingly, they often show preference for dry soils low in total organic C (69) and would be expected in the barren quarry soils of this study. The COVG treatment showed some of the highest proportions of *Actinobacteria*; however, it and COHort showed different enriched taxa: *Nocardioides* and two genera previously grouped into *Mycobacterium* (70) for COVG versus *Isoptericola* (some previously known as *Cellulosimicrobium*) (71) and *Nocardiopsis* for COHort, generally at the expense of *Streptomyces*, which remained most dominant in controls. This is perhaps consistent with the observations of some of the former genera being faster growing and/or isolated from richer soil types (72–75), potentially allowing them to outcompete *Streptomyces* in the organically richer environments of the amended soils. The most enriched genus in the plant-based amendments (COVG and COHort) was the functionally complex and plant-growth-promoting *Paenibacillus* (76, 77), being nearly absent from control soils and SS. It was again nearly missed by the 16S data (<1% summed across all samples) and could be an example of a taxon that thrived in the study plots after possibly being introduced from the original amendment, due to multiple reports of this *Firmicutes* member being frequently isolated from composts (78–80). The plant-growth-promoting and disease-suppressive *Lysobacter* (*Gammaproteobacteria*) (81) was the most significantly enriched genus in SS for the MGS method and was detected in the 16S but was not seen as relevant to the community patterns in the latter. However, the gammaproteobacterium *Pseudomonas* and the *Bacteroidetes* members *Sphingobacterium*, *Pedobacter*, and *Flavobacterium* were consistent between methods as being significant contributors to the sewage amendment patterns. That being said, a

multitude of other genera identified as being key players in the 16S data were determined by MGS either to be nonsignificant, such as *Novosphingobium* and *Staphylococcus*, or to compose a minor (<1%) proportion of the soils, such as *Gemmatimonas* and *Taibaiella* (Fig. 4). Overall, the MGS missed only one genus (*Verticia*) found to be important in the 16S, whereas the 16S did not detect 9 genera seen as differentially abundant in the MGS, and a further 12 genera were in such low proportions in the 16S that they were ignored.

**Functional redundancy across soils with potential small shifts in enzymes and pathways.** Although one can utilize predictive software, such as PICRUSt2 (82), to get a proxy of function from 16S data, shotgun MGS is still the "gold standard" for the direct analysis of microbial community function in the environment—although this still remains "predicted" function versus "expressed" function from metatranscriptomics. That being said, MGS data still present substantial challenges in accurate assignment, visualization, and interpretation, especially for soil environments given their higher complexity (52). Our MGS depth was not enough to consider sequence binning and assembly, but those techniques also have significant caveats that are still under examination (83); therefore, we restricted our analysis to the more straightforward description of assigned functions and their groupings into pathways. Similar to the taxonomic data derived from the MGS above, the functional data also showed distinct separation of each treatment type, regardless of whether individual EC distributions or combined pathways were examined, indicating that the amendments were responsible for pushing the community functional profiles away from their original control soil values. Although these patterns were still very clear, they were not driven by vast differences between treatments. In contrast to the taxonomic data where substantial swings in presence/proportions of individual phyla or genera could be seen between soils, the functions showed substantial redundancy in that most functions were present in all soils and showed only relatively minor proportional differences. The result was that substantially fewer significant features were detected (many times none at all, especially for pathways) and represented true redundancy, not being simply the product of the high feature number increasing the multiple test burden, as the total number of individual functions annotated (see Data Set S4 in the supplemental material) was a similar order of magnitude (~3,000) as the total number of genera. Functional redundancy is commonly observed at the level of broad-scale ecosystem functions or individual pathways in soils and other environments (9, 84, 85) and is considered a natural emergent property of all microbial systems (86).

Despite the functional redundancy "dampening" the major signals in the MGS data, various modest conclusions can still be drawn about the overall community functioning of the control and treatment restoration plots. The MetaCyc results showing slightly higher total biosynthesis and degradation mapping in all amended soils than in controls accords with higher nutrient levels (total organic C, total N, and assimilable P), higher respiration rates, increased enzyme activities, and 10- to 40-fold increases in bacterial biomass (via fatty acid methyl ester [FAME] analysis) observed in the treated soils as part of the companion studies on the quarry restoration project (16, 26, 30). Similarly, the overall MetaCyc mapping order of SS > (COVG/COHort) > control matches the pattern of increased organic matter remineralization in all treatments over control soils, with the highest in sewage, followed by COVG and COHort at intermediate levels (16). In terms of the few specific enzyme activities previously measured in the study samples using biochemical techniques (as standard proxies for metabolic activity) (16), their patterns did not consistently match MGS read numbers mapped to those functions (Fig. S6). Urease read numbers were highest for control soils versus all other treatments, whereas enzyme activity was measured as highest in COHort. $\beta$-Glucosidase read mapping was not significantly different while showing the typical trend in activity (SS > COHort > COVG > control). Mapping to alkaline phosphatase followed a pattern of SS > control > (COVG/COHort), whereas activity also had SS as highest, but COHort and COVG were still well above control levels. However, the pattern for total dehydrogenase category mapping (all EC 1.1.1.X enzymes) did match the exact activity pattern of SS > COHort > COVG > control. The disconnect between

biochemical activities and gene mapping could highlight specificity problems in the chemical assays or, more likely, represents the difference between conducting metatranscriptomics (i.e., gene expression) and conducting MGS in this study, which catalogues the changes in gene copies encoded by the genomes of the community, whether active or not (53). It is at times difficult to tease apart whether increases in reads to specific functions in MGS data are simply a result of "piggybacking," where a function increases simply because it is in the genomes of organisms increasing, or whether those functions are truly selected for in the samples. An example of the former was possibly the increase of the osmolyte ectoine in COHort, known to be present in *Paenibacillus* (87), concomitant with the increase of that same genus in COHort. However, our data also contained examples of the latter—functions and pathways that seemed to have logical selection patterns implying actual use/activity: (i) the near-absence of mapping to denitrification pathways in the N-starved control soils compared to the N-rich amendments and (ii) the inverse pattern of increased mapping to glycogen pathways (used for energy storage during nutrient stress) (88) in the nutrient-poor control soils compared to most other carbohydrate biosynthesis categories showing increased mapping to all amendments. Having metatranscriptomic data for these soil treatments may have definitively resolved some of these open questions. However, sampling in the field for RNA analysis, which requires snap-freezing samples in liquid N immediately, is often challenging, and few soil metatranscriptomes have been yet completed (89). The assumption behind the much more often-used MGS is that if functions are under selection in particular environments/treatments, then the genomes carrying those functions will also be positively selected, resulting in trends still visible in MGS data, even if obscured somewhat by inactive organisms that are still sequenced.

One caveat of our study data is that the functional differences here do need to be taken in context, as some of the analysis methods used (either ALDEx2 or LEfSe, as the case may be) did not find any significant differences at all for most of the functional and pathway comparisons. This seems to imply that even very minor functional shifts, connected to a small number of taxon changes observed above, are capable of vastly improving treated soil performance. Either this is the case, or shared functions (redundancy) across untreated and treated plots are simply being expressed to much greater degrees in the treated soil which, as mentioned above, could be further elucidated by metatranscriptomics.

**Conclusions.** Overall, the metagenomic analysis displayed more functional redundancy and overall similarity between the various soils than perhaps was expected at the outset. It has recently been shown that arid ecosystem metagenomes appear to have higher proportions of genes that are unknown in reference databases (51), potentially leading to difficulties in annotating and, therefore, discriminating arid soil samples from temperate soil samples. However, the extracted taxonomic identities of the reads, and the specific functions encoded by them, in our quarry soils were more than sufficient to discriminate the various treatments and showed significant impacts of the restorations on community profiles. These significant shifts mirrored the overall substantive soil quality and vegetation-supporting improvements that occurred in the treated plots versus the control plots (16, 26), the latter of which were much poorer overall than the non-human-impacted natural soils used as a reference here. Subsequent study of these restorations using metatranscriptomics might help further demonstrate that the patterns of gene abundances here are indeed supported by actual expression changes in the actively growing segment of the bacterial population, which may also lead to substantially more separation between the treatment and control plots. We are also mining the existing data for more information on specific biogeochemical pathways and attempting to reconstruct further taxon-function relationships that were beyond the scope of this first broad profiling of the communities.

## MATERIALS AND METHODS

**Study area.** The study was carried out in completely degraded soils from a limestone quarry located in the Gádor mountain range in Almería (southeast Spain; 36°55′18″ N, 02°30′40″ W). The geological

material was mainly formed by limestones and dolomites with calcareous sandstones, and marly and loamy marls forming Regosols (90). This region has a semiarid Mediterranean climate with an average annual rainfall of 242 mm; most of these events are in winter and autumn. Potential evapotranspiration is 1,225 mm year$^{-1}$. Summers are hot and dry with maximum temperatures recorded in August of 31°C and minimum temperatures of approximately 8°C in January (19). Native vegetation is composed mainly of *Macrochloa tenacissima* (L.) Kunth (= *Stipa tenacissima* L.), accompanied by small shrubs such as *Ulex parviflorus* Pourr. and *Anthyllis cytisoides* L., as well as dispersed individuals of *Maytenus senegalensis* (Lam.) Exell., *Pistacia lentiscus* L., and *Rhamnus lycioides* L. (14).

**Experimental design and soil sampling.** The experimental plots were established in a totally exploited flat site in the quarry at 362 m above sea level (masl). See the work of Soria et al. (26, 30) for pictures and detailed descriptions of the experimental setup. Briefly, the marl substrate was homogenized and decompacted using heavy machinery from the quarry (i.e., mechanical excavators and bulldozers) to decrease erosion by rainfall events and facilitate soil infiltration. Eighteen experimental plots of 50 m$^2$ each (10 m by 5 m) were then demarcated, and five restoration treatments consisting of different organic amendments from wastes of different origin and chemical composition were applied randomly to these experimental plots using a mechanical backhoe also available in the quarry facilities. These treatments increased the organic carbon content over 3% in the first 20 cm of depth. The organic amendments applied were (i) 100% vegetable compost manufactured from garden waste (COVG); (ii) 100% vegetable compost manufactured from horticultural greenhouse crop waste (COHort); (iii) sewage sludge waste treated by mesophilic digestion, thermal dehydration at 70°C, and centrifugation (SS); (iv) an amendment made from a 50:50 mixture of COVG + SS; and (v) an amendment made from a 50:50 mixture of COHort + SS. In addition, unamended experimental plots were used as control plots (Control). Finally, the experimental design had 3 plots per each treatment (i.e., 3 replicates) × 5 different restoration treatments plus the controls = 18 experimental plots total. Moreover, surrounding natural soils close to the experimental plots, but not disturbed by mining activities, were chosen as quality reference soils (Natural) (91).

Once the treatments were applied, two different species of native plants characteristic of the study area were selected for the restoration: 40 plants of *Macrochloa tenacissima* L. Kunth and 10 plants of *Olea europaea* L. var. *sylvestris* Brot. were planted 1 m apart in each experimental plot (restored and control soils). Irrigation was established during the first summer just after planting to ensure the vegetation's survival because of the harsh climatic conditions typical of the Mediterranean semiarid area, such us long summer droughts and high temperatures (19).

Composite soil samples, from mixing 10 random subsamples, were collected up to a depth of 10 cm throughout each experimental plot in the gaps between plants after 6 months of application of the organic amendments to study the short-term microbial responses. A total of 21 soil samples (3 replicates × 5 restoration treatments = 15, plus 3 replicates from unamended control experimental plots and 3 replicates from surrounding natural soils) were taken to the laboratory in isothermal bags. In the laboratory, the samples were sieved through a 2-mm screen and preserved at −20°C for DNA extraction and next-generation sequencing.

**DNA extraction, metagenomic library preparation, and sequencing.** The DNA contained in 0.3 g of soil was extracted from the each of the 21 samples using the DNeasy PowerSoil kit (Qiagen, Hilden, Germany) and later quantified with an ND-2000 Nanodrop spectrophotometer (Thermo Fisher Scientific, USA). Approximately 5 ng of extracted DNA was then used as input for the Illumina Nextera DNA Flex library prep kit and barcoded using the Nextera DNA CD indexes, per the manufacturer's instructions. Final libraries were quantified using the Invitrogen Quant-iT double-stranded DNA (dsDNA) (high-sensitivity) assay using a microplate reader, and equal amounts of each library were pooled and then sequenced at the Integrated Microbiome Resource (IMR; Dalhousie University) to an average read depth of 5 million 2 × 150-bp paired-end (PE) reads on an Illumina NextSeq 550 using the High Output v2.0 chemistry.

**Metagenomics data taxonomic and functional annotations.** Raw FASTQ files were demultiplexed on instrument and then processed using a pipeline under development at the IMR as part of the continually evolving Microbiome Helper (92) repository, the current version of which is available at https://github .com/LangilleLab/microbiome_helper/wiki/Metagenomics-Standard-Operating-Procedure-v3. In summary, raw reads were quality controlled using KneadData v0.7.2 (https://github.com/biobakery/kneaddata), which employs Trimmomatic v0.36 (93) (options: SLIDINGWINDOW:4:20 MINLEN:50) and Bowtie2 v2.2.3 (94) (options: –very-sensitive –dovetail) to filter low-quality reads and screen out potentially contaminant sequences against the human (GRCh38) and phiX174 genomes. A summary of raw and post-quality-control (post-QC) read numbers per sample, along with the key to which group they belong, is available as Data Set S1 in the supplemental material. "Raw" taxonomic composition was determined using Kraken2 v2.0.8 (95) (option: –confidence 0.1), based upon a 150-mer database built from the entire NCBI RefSeq Complete v93 database, and final taxonomic abundance profiles were generated using Bracken v2.0 (96) (option: -t 10). Custom PERL and Python scripts (available above) were then used to do functional mapping of reads using MMseqs2 (97) against the entire UniRef90 database (http://www.uniprot.org/uniref), select the top hit for each read, map functions to Enzyme Commission (EC) numbers, and then generate either unstratified (function-only) or stratified (linking the above Kraken2 taxonomy) matrices. Two PICRUSt2 v2.2.0-b scripts (82) were used to append functional descriptions (*add_descriptions.py*) to the above matrices and to generate MetaCyc pathway coverages (*pathway_pipeline.py*). Final functional coverages were normalized to reads per kilobase per million (RPKM). Processes were parallelized using GNU parallel (http://www.gnu.org/software/parallel/).

**Statistical analyses and visualizations.** Taxonomic, functional, and pathway matrices were imported into QIIME2 v2020.08 in order to generate ordination plots, conduct differential abundance testing for

significant taxa/functions/pathways, and prepare heatmaps. The input tables were normalized and low-abundance filtered (to remove signal noise constituting <1% of cumulative counts across all 21 samples) as follows: taxonomy to 1 million reads per sample with a minimum of 10,000 per genus, functions to 85,000 RPKM per sample with a minimum of 850 per EC number, and pathway abundance to 8,500 RPKM per sample with a minimum of 85 per pathway. Principal-coordinate analyses (PCoAs) from the Bray-Curtis distance matrices, ADONIS tests of the groupings, and Shannon's $H$ were created using the *qiime diversity* functions. Differential abundance testing was conducted by two methods: (i) using the ALDEx2 (98) plugin available for QIIME v2019.7 (https://library.qiime2.org/plugins/q2-aldex2/24/) and (ii) running LEfSe (99) on the Huttenhower Galaxy server (https://huttenhower.sph.harvard.edu/galaxy/). Nonparametric base statistics, such as richness and diversity comparisons (e.g., Kruskal-Wallis), were conducted in the PAST v4.03 statistical program (100). Where already not integrated into the test, the Benjamini-Hochberg false-discovery rate was systematically applied to correct for multiple testing. Further visualizations were done by importing the unstratified functional matrix as a SmartTable into the MetaCyc Omics Dashboard (101) (https://metacyc.org/dashboard/dashboard-intro.shtml); however, the complexity of the matrix was reduced by excluding trace functions that did not represent at least 100 RPKM summed across all 21 samples.

**Data availability.** The raw metagenomic sequences presented in this study are available at the ENA under accession number PRJEB47869. The full taxonomic profiles at the phylum and genus levels are available as Data Sets S2 and S3, and the full unstratified functional profiles are available as Data Set S4.

## SUPPLEMENTAL MATERIAL

Supplemental material is available online only.

**FIG S1**, PDF file, 0.03 MB.
**FIG S2**, PDF file, 0.04 MB.
**FIG S3**, PDF file, 0.05 MB.
**FIG S4**, PDF file, 0.1 MB.
**FIG S5**, PDF file, 0.1 MB.
**FIG S6**, PDF file, 0.1 MB.
**DATA SET S1**, XLSX file, 0.01 MB.
**DATA SET S2**, TXT file, 0.02 MB.
**DATA SET S3**, TXT file, 0.7 MB.
**DATA SET S4**, TXT file, 0.5 MB.

## ACKNOWLEDGMENTS

This work was supported by the Spanish Ministry of Economy and Competitiveness (MINECO), and FEDER funds, through the project CGL2017-88734-R (BIORESOC) MINECO/AEI/FEDER, UE and FEDER-Junta de Andalucía Research Projects RESTAGRO (UAL18-RNM-A021-B), and Restoration of Abandoned Agricultural Soils in Semiarid Zones to Improve Productivity and Soil Quality and Enhance Carbon Sequestration (P18-RT-4112). Isabel Miralles is grateful for funding received from the Ramón y Cajal Research Grant (RYC-2016-21191) from the Spanish Ministry of Economy, Industry and Competitiveness (MINECO). Raúl Ortega thanks his postdoctoral contract HIPATIA of the University of Almería Research Plan.

We declare that we have no competing interests.

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
