## [Reviewer comments · mSystems]

Functional and Taxonomic Effects of Organic Amendments on the Restoration of Semiarid Quarry Soils

Isabel Miralles, Raúl Ortega, and Andre Comeau

Corresponding Author(s): Isabel Miralles, University of Almeria

Review Timeline:

Submission Date:	June 14, 2021
Editorial Decision:	July 16, 2021
Revision Received:	October 8, 2021
Editorial Decision:	November 1, 2021
Revision Received:	November 3, 2021
Accepted:	November 5, 2021

Editor: Nick Bouskill

Reviewer(s): Disclosure of reviewer identity is with reference to reviewer comments included in decision letter(s). The following individuals involved in review of your submission have agreed to reveal their identity: Daniel Rath (Reviewer #1); Priyanka Kushwaha (Reviewer #2)

Transaction Report:

DOI: <https://doi.org/10.1128/mSystems.00752-21>

July 16, 2021

Dr. Isabel Miralles
University of Almeria
Department of Agronomy & Center for Intensive Mediterranean Agrosystems and Agri-food Biotechnology (CIAIMBITAL)
Almeria
Spain

Re: mSystems00752-21 (Functional and Taxonomic Effects of Organic Amendments on the Restoration of Semiarid Quarry Soils)

Dear Dr. Isabel Miralles:

Thank you for submitting your manuscript to mSystems. Your manuscript has been reviewed by two experts in the field. Both noted the manuscript is well written and surmises an extremely interesting dataset. However, as you will see in the reviews below, both authors identified areas that must be accounted for before the manuscript can be considered for publication. In particular, both reviewers raised questions about the statistical approaches, the amount of important data within the supplemental material, and the strength of the hypotheses motivating the work. I have also read through the manuscript and agree with the reviewer's assessments.

Please also note, mSystems also requires that data be made accessible prior to acceptance. The 'pending' data accessibility statement will need to be updated with either a DOI from a public-facing data repository or an accession number.

To this end, I am returning your manuscript for major revision prior to resubmission. Below you will find instructions from the mSystems editorial office and comments generated during the review.

Preparing Revision Guidelines

For complete guidelines on revision requirements for your article type, please see the journal Article Types requirement at <https://journals.asm.org/journal/mSystems/article-types>. **Submissions of a paper that does not conform to mSystems guidelines will delay acceptance of your manuscript.**

Sincerely,

Nick Bouskill

Editor, mSystems

Journals Department
Reviewer comments:

Reviewer #1 (Comments for the Author):

This paper is an interesting look at the taxonomic and functional changes of the soil microbiome that occur following 6 months of organic amendments and native plant introduction in mine restoration sites. Authors found significant differences between treatments in the overall taxonomic profiles, but fewer significant differences when looking at overall functional profiles. The study did note significant differences in specific functional gene categories between individual treatments and the combination of all other treatments, specifically in C degradation and N transformation.

The data and conclusions presented here are novel and of interest to the wider field of soil microbiology, and by including both a control and an undisturbed treatment in the experiment, the authors provide an excellent set of references to compare the effects of the amendments. The writing is easy to understand with the exception of some run-on sentences, and the authors contribute to the wider conversation in microbiome research by using their data to discuss 16S primer bias, potential vs actual functions, and functional redundancy in the soil microbiome.

However, there are several issues with the analysis and presentation of the data.

It is not clear what the comparison of each single treatment to the combination of all other treatments in the functional analysis is meant to convey. How is this data to be interpreted in the context of the paper's goals - i.e. how the amendments have changed taxonomy and function relative to the control or undisturbed soils?

This comparison is made more confusing by the fact that the authors do not discuss any of the functional data they included in the results (lines 196-232), instead using Supplementary Graphs S8-S10 to support their argument. If these S8-S10 graphs are to be the main support for the "Functional redundancy" section of the discussion, they should be included in the results.

What is the difference between control and natural plots? In the methods section, authors state that control plots are "unamended experimental plots", indicating that these soils were impacted by mining. The natural plots close to the experimental plots were undisturbed by mining. It then seems strange that control and natural plots would show zero significantly different genera - does this mean that mining and topsoil removal had no effect on the soil microbial community? Was this the same for functional analysis? This is a very important point that should be addressed in the paper

The presentation and color scheme for Figures 3, 5, and 6 are difficult to interpret. The different color schemes for negative and positive differences in mean proportions are confusing, as all the values are relative to the control (eg. it is not clear what the difference is for a blue dot vs a grey dot in Fig 3A, as we can already see whether the value is positive or negative). Authors should consider making the dots a single color, or using colors that do not overlap with the treatment labels to avoid confusion.

Authors need to expand a bit more on the ordination and statistical tests used in presenting the data. What type of ordination was used? (I think PCA?) Also, authors should reference what statistical test they used in the ordination to determine whether communities were significantly different (ANOVA?), as from the manuscript it just appears that the grouping was visually compared.

Was the data transformed or not? The shape of the plots in Fig 4 may denote the presence of a horseshoe effect, where the second axis is an arched function of the first axis, indicating that authors may need to consider DCA (detrended correspondence analysis) depending on what type of ordination they originally did. As a side note, given that the authors have environmental data for these samples, it would be interesting to know what gradients each of these axes align with, if the test was indeed a PCA.

Authors should introduce some of the caveats of shotgun sequencing (MGS) in the introduction: i.e. that functional gene presence denotes potential, not actual functional differences; and the potential differences between 16S and MGS. Since these concepts are used to explain a lot of the trends in their data, priming readers to talk about them in the introduction may be helpful.

The hypothesis at the end of the introduction seems vague - were there more targeted hypotheses that could be supported by the data e.g. SS will increase CAZyme potential?

I think that this paper is a fascinating look into amendment-induced changes in the soil microbiome, and even though the authors state that it is a preliminary paper, there is still some additional analysis that should be done. A more detailed comparison of the difference between 16S and MGS results is warranted - were the samples taken at the same time of year/same year? Were they treated differently in storage? While primer bias is a likely culprit to explain the differences, there are other lines of support

that can be brought in to strengthen this argument.

If environmental data is available, including it with the ordination can provide valuable insights into why exactly the overall taxonomic profiles were different. I would also encourage authors to embrace the potential lack of differences in overall functional analyses- perhaps this is an indication that the restoration treatments do not result in large overall changes to the microbial community profiles, and instead only affect very specific taxa and functions. Given that the authors already have measurements of how these soils respond differently in their previous 2021 paper, it may be that relatively small taxonomic and functional gene changes are all that are needed to improve soil function.

Overall, I think this is a fascinating dataset with the potential to be a great contribution to the discussion around soil metagenomes and human activities, but further analysis and interpretation of results is needed to achieve this potential.

Other issues are listed below:

Authors should provide a short summary of what the abbreviations for the different treatment codes mean (COHort, COVG, SS) when they are introduced so that readers do not need to skip ahead to the methods section to find out.

What do authors mean by "metagenome bias" (lines 38, 251)? Introducing this concept in the introduction with some supporting references could help to clarify.

Authors reference "introduced microbes" (317-319) - please cite sources to support the concept that microbes introduced in compost can persist in soil environments

What do authors mean by "The two treatment mixtures...displayed exactly intermediate patterns" (Line 147)? Is this statement based on some type of distance-based ordination (NMDS, PO, PCoA) as opposed to an eigenanalysis-based ordination (PCA, CCA)? The underlying philosophy of the eigenanalysis-based methods is fundamentally different from distance-based methods: eigenanalysis based methods attempt to faithfully place species along gradients (either inferred or directly related to measured variables), and not to faithfully relate difference to distance.

Reviewer #2 (Comments for the Author):

In this manuscript, Miralles et al. aimed to evaluate the functional and taxonomic effects of organic amendments on the restoration of semiarid quarry soils using metagenomics (MGS) analysis. They compare the microbial taxonomic composition using two techniques 1) 16S rDNA amplicon sequencing (previously published) and 2) MGS in this study for different organic amendments applied to the quarry soils along with the controls. Further, they evaluate the functional profiles of the different soils. As authors have mentioned in their study, soil MGS is complex and only a handful of studies have used this methodology compared to 16S rDNA amplicon sequencing. Therefore, this study adds to the limited knowledge of soil metagenomics, particularly in mining soils. Overall, this manuscript was well written. However, there are some concerns that are listed for authors to improve the study.

General points:

1. The authors should note that there are several published studies from mining soils in arid soils that have assessed the role of organic amendments in revegetation of mining soils (e.g., The Maier papers from Arizona, US). Note also that Chen et al. 2020 (cited later as Ref 85) study on arid soil metagenomes is not cited in Lines 121-124.
2. The ordination plots are not supported by any statistical tests. For e.g., ordination plots are supported by running ANOSIM or PERMANOVA to evaluate if treatment type significantly affects the clustering of samples within the treatment. This is missing for taxonomic as well as functional ordination plots.
3. Although authors compare the results from previously published 16S paper, they do not have any comparative analysis included in the main manuscript or supplementary material.
4. The tables and figures are repetitive in the way the data are represented. This makes it harder to understand what different data are being represented in 3 tables that look similar.
For tables 1-3, Venn-Diagrams may be a better way of visually representing these data. For figures 3, 5 & 6: Heatmaps could be another way of representing these data.

Line specific comments:

Lines 141-143: As the methods are in the end, please define the treatments here.

Line 172-174: Which statistical tests are the authors referring to when they use the term "Differential analysis of these genera"? Student's t-test or ANOVA?

Line 193:195: Why are rich data not shown? Richness data could be included in the supplementary material.

Lines 825-826: Does that mean that taxonomic associations other than genera were not considered for Fig 1 ordination?

Line 834: Is the "difference in mean" referring to the "difference in mean proportions or abundance" of taxa?

Lines 830, 836, 851, 860: How were the differentially abundant taxa and function calculated? Are they significantly abundant?

On the methodology:

Lines 452-453: The authors state that 40 plants of *Macrochloa tenacissima* L. Kunth and 10 plants of *Olea europaea* L. var *sylvestris* Brot. were planted in each plot. However, the authors do not specify if the soil sample was collected under the canopy of the plants or in open spaces/gaps between plants.

Line 464: Was the total genomic DNA fragmented during the Illumina Nextera DNA Flex Library Prep?

Line 468: Was 5 million the average number of reads in each sample?

Lines 470-490: Metagenomics analysis: 1) How was the metagenome-assembled?;

2) Was there a cut-off to remove reads with shorter base pairs ?;

3) What was the total number of reads per sample after trimming and filtering? A table with these numbers would be good in the supplementary material.

Statistical analyses: Authors have compared different treatments for taxonomic and functional differences. However, the significance values of these comparisons are not listed in the results. In addition, Student's t-test and ANOVA are not the best tests to use for these comparisons. Tools such as LEfSe or DESeq2 would allow the authors to show which taxa/functional pathways are over-represented or enriched in one treatment over the other. These tools are statistically more robust.

Minor comment:

Line 391: is "more oft-used MGS" referring to "more often-used MGS"?

Review - mSystems

This paper is an interesting look at the taxonomic and functional changes of the soil microbiome that occur following 6 months of organic amendments and native plant introduction in mine restoration sites. Authors found significant differences between treatments in the overall taxonomic profiles, but fewer significant differences when looking at overall functional profiles. The study did note significant differences in specific functional gene categories between individual treatments and the combination of all other treatments, specifically in C degradation and N transformation.

The data and conclusions presented here are novel and of interest to the wider field of soil microbiology, and by including both a control and an undisturbed treatment in the experiment, the authors provide an excellent set of references to compare the effects of the amendments. The writing is easy to understand with the exception of some run-on sentences, and the authors contribute to the wider conversation in microbiome research by using their data to discuss 16S primer bias, potential vs actual functions, and functional redundancy in the soil microbiome.

However, there are several issues with the analysis and presentation of the data.

- 1) It is not clear what the comparison of each single treatment to the combination of all other treatments in the functional analysis is meant to convey. How is this data to be interpreted in the context of the paper's goals - i.e. how the amendments have changed taxonomy and function relative to the control or undisturbed soils?

This comparison is made more confusing by the fact that the authors do not discuss any of the functional data they included in the results (lines 196-232), instead using Supplementary Graphs S8-S10 to support their argument. If these S8-S10 graphs are to be the main support for the "Functional redundancy" section of the discussion, they should be included in the results.

- 2) What is the difference between control and natural plots? In the methods section, authors state that control plots are "unamended experimental plots", indicating that these soils were impacted by mining. The natural plots close to the experimental plots were undisturbed by mining. It then seems strange that control and natural plots would show zero significantly different genera - does this mean that mining and topsoil removal had no effect on the soil microbial community? Was this the same for functional analysis? This is a very important point that should be addressed in the paper
- 3) The presentation and color scheme for Figures 3, 5, and 6 are difficult to interpret. The different color schemes for negative and positive differences in mean proportions are confusing, as all the values are relative to the control (eg. it is not clear what the difference is for a blue dot vs a grey dot in Fig 3A, as we can already see whether the value is positive or negative). Authors should consider making the dots a single color, or using colors that do not overlap with the treatment labels to avoid confusion.

- 4) Authors need to expand a bit more on the ordination and statistical tests used in presenting the data. What type of ordination was used? (I think PCA?) Also, authors should reference what statistical test they used in the ordination to determine whether communities were significantly different (ANOVA?), as from the manuscript it just appears that the grouping was visually compared.

Was the data transformed or not? The shape of the plots in Fig 4 may denote the presence of a horseshoe effect, where the second axis is an arched function of the first axis, indicating that authors may need to consider DCA (detrended correspondence analysis) depending on what type of ordination they originally did. As a side note, given that the authors have environmental data for these samples, it would be interesting to know what gradients each of these axes align with, if the test was indeed a PCA.

- 5) Authors should introduce some of the caveats of shotgun sequencing (MGS) in the introduction: i.e. that functional gene presence denotes potential, not actual functional differences; and the potential differences between 16S and MGS. Since these concepts are used to explain a lot of the trends in their data, priming readers to talk about them in the introduction may be helpful.
- 6) The hypothesis at the end of the introduction seems vague - were there more targeted hypotheses that could be supported by the data e.g. SS will increase CAZyme potential?

I think that this paper is a fascinating look into amendment-induced changes in the soil microbiome, and even though the authors state that it is a preliminary paper, there is still some additional analysis that should be done. A more detailed comparison of the difference between 16S and MGS results is warranted - were the samples taken at the same time of year/same year? Were they treated differently in storage? While primer bias is a likely culprit to explain the differences, there are other lines of support that can be brought in to strengthen this argument.

If environmental data is available, including it with the ordination can provide valuable insights into why exactly the overall taxonomic profiles were different. I would also encourage authors to embrace the potential lack of differences in overall functional analyses- perhaps this is an indication that the restoration treatments do not result in large overall changes to the microbial community profiles, and instead only affect very specific taxa and functions. Given that the authors already have measurements of how these soils respond differently in their previous 2021 paper, it may be that relatively small taxonomic and functional gene changes are all that are needed to improve soil function.

Overall, I think this is a fascinating dataset with the potential to be a great contribution to the discussion around soil metagenomes and human activities, but further analysis and interpretation of results is needed to achieve this potential.

Other issues are listed below:

- 7) Authors should provide a short summary of what the abbreviations for the different treatment codes mean (COHort, COVG, SS) when they are introduced so that readers do not need to skip ahead to the methods section to find out.
- 8) What do authors mean by “metagenome bias” (lines 38, 251)? Introducing this concept in the introduction with some supporting references could help to clarify.
- 9) Authors reference “introduced microbes” (317-319) - please cite sources to support the concept that microbes introduced in compost can persist in soil environments
- 10) What do authors mean by “The two treatment mixtures...displayed exactly intermediate patterns” (Line 147)? Is this statement based on some type of distance-based ordination (NMDS, PO, PCoA) as opposed to an eigenanalysis-based ordination (PCA, CCA)? The underlying philosophy of the eigenanalysis-based methods is fundamentally different from distance-based methods: eigenanalysis based methods attempt to faithfully place species along gradients (either inferred or directly related to measured variables), and not to faithfully relate difference to distance.

Reviewer #1 (Comments for the Author):

This paper is an interesting look at the taxonomic and functional changes of the soil microbiome that occur following 6 months of organic amendments and native plant introduction in mine restoration sites. Authors found significant differences between treatments in the overall taxonomic profiles, but fewer significant differences when looking at overall functional profiles. The study did note significant differences in specific functional gene categories between individual treatments and the combination of all other treatments, specifically in C degradation and N transformation.

The data and conclusions presented here are novel and of interest to the wider field of soil microbiology, and by including both a control and an undisturbed treatment in the experiment, the authors provide an excellent set of references to compare the effects of the amendments. The writing is easy to understand with the exception of some run-on sentences, and the authors contribute to the wider conversation in microbiome research by using their data to discuss 16S primer bias, potential vs actual functions, and functional redundancy in the soil microbiome.

However, there are several issues with the analysis and presentation of the data.

It is not clear what the comparison of each single treatment to the combination of all other treatments in the functional analysis is meant to convey. How is this data to be interpreted in the context of the paper's goals - i.e. how the amendments have changed taxonomy and function relative to the control or undisturbed soils?

Agreed – these combinations led to increased complexity that was not necessary. The presentation of the data has been streamlined to always include the same categories/comparisons now: consistently just each treatment alone against the Control, or all treated plots combined against the untreated plots (= Naturals+Controls, since virtually identical). Terminology has also been updated to use “Treated vs Untreated” to make things clearer.

This comparison is made more confusing by the fact that the authors do not discuss any of the functional data they included in the results (lines 196-232), instead using Supplementary Graphs S8-S10 to support their argument. If these S8-S10 graphs are to be the main support for the "Functional redundancy" section of the discussion, they should be included in the results.

The Discussion of those results highlighted that, out of the 1000s of functions and 100s of pathways, that very few actually showed differences in any comparisons...and that these differences all disappeared when a very conservative “noise filter” of >0.1% was added – hence functional redundancy since the only differences came from minor/trace, statistically non-significant differences. The Results and Discussion sections have now been a bit modified due to the shift to LEfSe+ALDEx2, but continue to show only a handful of individual differences with very small effect sizes leading, again, to the conclusion of redundancy. The MetaCyc Omics Dashboard graphs continue to remain in the Supplemental, but we are open to eventually moving them into the main text if figure limits are not imposed. The interpretation of those “cumulative-type” graphs are a bit more ambiguous, hence why we originally left them in the Supplementary – they are sums across many different functions into broad categories and are more useful to see overall “trends” at a macro scale (akin to viewing taxa graphs at the phylum-level, where significant differences become obscured/minimized) and do not show many

major differences (and it is impossible to add error bars to them to see the overlap)...but this is often the case when looking at any samples at these high functional categories (as when one compares phylum graphs across samples/environments).

What is the difference between control and natural plots? In the methods section, authors state that control plots are "unamended experimental plots", indicating that these soils were impacted by mining. The natural plots close to the experimental plots were undisturbed by mining. It then seems strange that control and natural plots would show zero significantly different genera - does this mean that mining and topsoil removal had no effect on the soil microbial community? Was this the same for functional analysis? This is a very important point that should be addressed in the paper

Correct – the natural plots were nearby soil not having seen mining activity and the control plots were within the old mining zone, but unamended by composts (ie: true negative controls). It was also surprising to us that the natural soils were not more different and formed one of our (updated) hypotheses at the outset. This important point has now been formally discussed in a large new section (lines 261-290) of the Discussion. We hypothesize that the amount of time (and thinner topsoil nature of these arid soils) since the end of mining has been sufficient for the reseeding (from resident and dispersed bacteria) of the control soils, so that their profiles now resemble natural soils. However, it is also pointed out that, even if their metagenomics profiles appear similar, they have vastly different biological activities measured with non-molecular means showing that the control soils remain very poor compared to native soils. It is evident that a technique like metatranscriptomics will be required to conclusively show that it is not their resident bacteria that are different, but the expression and growth that are vastly different in natural soils compared to the poor degraded mining soils (and which are vastly improved by the addition of the treatments, thing we do see in our previous papers and the metagenomics taxonomy herein).

The presentation and color scheme for Figures 3, 5, and 6 are difficult to interpret. The different color schemes for negative and positive differences in mean proportions are confusing, as all the values are relative to the control (eg. it is not clear what the difference is for a blue dot vs a grey dot in Fig 3A, as we can already see whether the value is positive or negative). Authors should consider making the dots a single color, or using colors that do not overlap with the treatment labels to avoid confusion.

The original plots were standard "post-hoc/confidence interval" type plots where, if the dot was to the right, then it was enriched in the treatment (hence colored that way = colors matched all other figures throughout the ms) and if it was to the left, it was enriched in the controls (or "all other samples", given the case). The confidence intervals were simply another way to represent the histograms on the left of the plots (but include error bars for the replicates). However, the point is now moot as these confidence interval plots have all been replaced by the LEfSe LDA plots which the reviewer and general audience might find more straightforward/clear.

Authors need to expand a bit more on the ordination and statistical tests used in presenting the data. What type of ordination was used? (I think PCA?) Also, authors should reference what statistical test they used in the ordination to determine whether communities were significantly different (ANOVA?), as from the manuscript it just appears that the grouping was visually compared.

Was the data transformed or not? The shape of the plots in Fig 4 may denote the presence of a horseshoe effect, where the second axis is an arched function of the first axis, indicating that authors may need to consider DCA (detrended correspondence analysis) depending on what type of ordination they originally did. As a side note, given that the authors have environmental data for these samples, it would be interesting to know what gradients each of these axes align with, if the test was indeed a PCA.

The reviewer is correct that the previous versions were eigen-based PCAs, that have more susceptibility to arch/horseshoe effects, and have now been updated to PCoA Bray-Curtis distance-based plots throughout. That being said, overall patterns are still essentially the same (slightly less of a shifting for COhort+SS now) and the results of the ADONIS tests have now also been added to the figures, reinforcing the very significant treatment groupings.

Authors should introduce some of the caveats of shotgun sequencing (MGS) in the introduction: i.e. that functional gene presence denotes potential, not actual functional differences; and the potential differences between 16S and MGS. Since these concepts are used to explain a lot of the trends in their data, priming readers to talk about them in the introduction may be helpful.

This has now been done on lines 124-129.

The hypothesis at the end of the introduction seems vague - were there more targeted hypotheses that could be supported by the data e.g. SS will increase CAZyme potential?

The hypothesis section has been expanded to include: 1) the anticipated (but incorrect) expectation of natural soils showing stark differences with control mining soils; and 2) the anticipated higher functional diversity/activity of the SS-treated soils from previous biological activity observations.

I think that this paper is a fascinating look into amendment-induced changes in the soil microbiome, and even though the authors state that it is a preliminary paper, there is still some additional analysis that should be done. A more detailed comparison of the difference between 16S and MGS results is warranted - were the samples taken at the same time of year/same year? Were they treated differently in storage? While primer bias is a likely culprit to explain the differences, there are other lines of support that can be brought in to strengthen this argument.

Agreed – as both reviewers wanted to see more substantial 16S comparison data, this has now been made a more significant part of the main manuscript with the inclusion of the new Fig.4 and some updated text changes. The exact same DNAs were used for both the 16S and metagenomics study herein, so no collection/timing, nor storage differences, and this further reinforces that the sole difference is the molecular method used (hence primer bias being the major culprit).

If environmental data is available, including it with the ordination can provide valuable insights into why exactly the overall taxonomic profiles were different. I would also encourage authors to embrace the potential lack of differences in overall functional analyses- perhaps this is an indication that the restoration treatments do not result in large overall changes to the microbial community profiles, and instead only affect very specific taxa and functions. Given that the authors already have measurements of how these soils respond differently in their previous 2021 paper, it may be that relatively small taxonomic and functional gene changes are all that are needed to improve soil function.

The restoration treatments do result in hundreds of taxonomic changes, some quite substantial, so we disagree with the reviewer on that aspect. However, we do agree with the reviewer that the minor functional differences observed do seem to be all that is required to make very substantial physicochemical and vegetation-supporting changes (ie: soil “function”) in the treated plots. We have more explicitly “embraced” these aspects by tweaking of various passages throughout and a more explicit section in the Discussion on lines 432-439.

Other issues are listed below:

Authors should provide a short summary of what the abbreviations for the different treatment codes mean (COHort, COVG, SS) when they are introduced so that readers do not need to skip ahead to the methods section to find out.

Agreed – this has been added to the end of the Intro as requested by both reviewers.

What do authors mean by "metagenome bias" (lines 38, 251)? Introducing this concept in the introduction with some supporting references could help to clarify.

The reference at both those lines is saying “less-biased metagenomics” (not “metagenome bias”) – this is the well-known principle that metagenomic analysis of community DNA is generally considered to suffer from less bias than 16S amplicon PCR (due to the exclusion of selective primers/PCR) – this has been elaborated upon, as mentioned above, in the Intro now (lines 124-129).

Authors reference "introduced microbes" (317-319) - please cite sources to support the concept that microbes introduced in compost can persist in soil environments

As the topic of microbial persistence in composts appears to be unproven either way in the small amount of available literature, we have modified the sentence to say it “...could be an example of a taxon that thrived in the study plots after possibly being introduced from the original amendment...”.

What do authors mean by "The two treatment mixtures...displayed exactly intermediate patterns" (Line 147)? Is this statement based on some type of distance-based ordination (NMDS, PO, PCoA) as opposed to an eigenanalysis-based ordination (PCA, CCA)? The underlying philosophy of the eigenanalysis-based methods is fundamentally different from distance-based methods: eigenanalysis based methods

attempt to faithfully place species along gradients (either inferred or directly related to measured variables), and not to faithfully relate difference to distance.

As mentioned above, the original eigen-based plots have now been replaced by distance-based PCoAs – the highlighted sentence has now been modified for the slight change in relative placement of COHort+SS (but now are distance, so similar language).

Reviewer #2 (Comments for the Author):

In this manuscript, Miralles et al. aimed to evaluate the functional and taxonomic effects of organic amendments on the restoration of semiarid quarry soils using metagenomics (MGS) analysis. They compare the microbial taxonomic composition using two techniques 1) 16S rDNA amplicon sequencing (previously published) and 2) MGS in this study for different organic amendments applied to the quarry soils along with the controls. Further, they evaluate the functional profiles of the different soils. As authors have mentioned in their study, soil MGS is complex and only a handful of studies have used this methodology compared to 16S rDNA amplicon sequencing. Therefore, this study adds to the limited knowledge of soil metagenomics, particularly in mining soils. Overall, this manuscript was well written. However, there are some concerns that are listed for authors to improve the study.

General points:

1. The authors should note that there are several published studies from mining soils in arid soils that have assessed the role of organic amendments in revegetation of mining soils (e.g., The Maier papers from Arizona, US). Note also that Chen et al. 2020 (cited later as Ref 85) study on arid soil metagenomes is not cited in Lines 121-124.

The Chen et al. 2020 reference has been added to line 124 (note reference # have now changed/shifted from original). The majority of the Maier papers (we already had referenced two regarding arid bacteria life histories) deal with restorations in very acidic mine tailings (quite a different physiological context) and are 16S-based (not metagenomic), however we have added them as the examples of 16S studies in human-impacted soils (lines 119-121).

2. The ordination plots are not supported by any statistical tests. For e.g., ordination plots are supported by running ANOSIM or PERMANOVA to evaluate if treatment type significantly affects the clustering of samples within the treatment. This is missing for taxonomic as well as functional ordination plots.

As mentioned above, the results of the ADONIS (PERMANOVA) tests have now also been added to the figures, reinforcing the very significant treatment groupings.

3. Although authors compare the results from previously published 16S paper, they do not have any comparative analysis included in the main manuscript or supplementary material.

Agreed – as both reviewers wanted to see more substantial comparison data, this has now been made a more significant part of the main manuscript with the inclusion of the new Fig.4 and some updated text changes.

4. The tables and figures are repetitive in the way the data are represented. This makes it harder to understand what different data are being represented in 3 tables that look similar.

For tables 1-3, Venn-Diagrams may be a better way of visually representing these data. For figures 3, 5 & 6: Heatmaps could be another way of representing these data.

The three tables have now been eliminated, but could not be represented by Venn diagrams as those are for highlighting shared things between items, whereas the significant features are counts specific to one sample or another. The equivalent data from LEfSe+ALDEx2 are now at the bottom of the new Fig.2 and original Figs.3+5+6 have been replaced with the LEfSe histograms. Heatmaps have been added to the new Fig.4 + the pathway figure (now Fig.7).

Line specific comments:

Lines 141-143: As the methods are in the end, please define the treatments here.

Agreed – this has been added to the end of the Intro as requested by both reviewers.

Line 172-174: Which statistical tests are the authors referring to when they use the term "Differential analysis of these genera"? Student's t-test or ANOVA?

These were originally two-group comparisons using the Welch's unequal variance t-test in STAMP, as was explicitly defined in the Methods. However, these have now all been replaced by the LEfSe and ALDEx2 results throughout.

Line 193:195: Why are rich data not shown? Richness data could be included in the supplementary material.

The taxonomic and functional richness + evenness data have now been added as the new Figure 2.

Lines 825-826: Does that mean that taxonomic associations other than genera were not considered for Fig 1 ordination?

We have the data for all levels, but the genus level is typically selected for this type of representation as it is a compromise between wanting to use the most detailed information possible (which would

normally be species/strain-level) and recognizing that the accuracy of taxonomic assignments from metagenomic data (and 16S) beyond the genus level are generally not very good due to database coverage issues. Using the abundance data from higher levels, such as Family or Phylum, tend to erase differences between most ecological samples, as higher levels are much more similar across geographic gradients/experiments and hence you are losing information using such high levels.

Line 834: Is the "difference in mean" referring to the "difference in mean proportions or abundance" of taxa?

The differences were as indicated on the axes of those figures ("difference in mean proportions"), but we have now replaced those figures with the new LEfSe ones.

Lines 830, 836, 851, 860: How were the differentially abundant taxa and function calculated? Are they significantly abundant?

Significant FDR-adjusted p-values were indicated right in the legend in order to indicate that we were defining "differently abundant" as having to be significantly so. Secondly, as mentioned above, these were originally two-group comparisons using the Welch's unequal variance t-test in STAMP, as was explicitly defined in the Methods. However, these have now all been replaced by the LEfSe and ALDEx2 results throughout.

On the methodology:

Lines 452-453: The authors state that 40 plants of *Macrochloa tenacissima* L. Kunth and 10 plants of *Olea europaea* L. var *sylvestris* Brot. were planted in each plot. However, the authors do not specify if the soil sample was collected under the canopy of the plants or in open spaces/gaps between plants.

The samples were taken in the gaps between plants and this has now been indicated.

Line 464: Was the total genomic DNA fragmented during the Illumina Nextera DNA Flex Library Prep?

The DNA is not physically fragmented, but the Nextera Flex method uses a transposase to cut the DNA into smaller fragments for sequencing – this is inherent to how the Illumina method works and no deviations were taken, hence done "as per the manufacturer's instructions".

Line 468: Was 5 million the average number of reads in each sample?

Line 468 explicitly stated "to an average read depth of 5 million" (see further comment below on Dataset S1).

Lines 470-490: Metagenomics analysis:

1) How was the metagenome-assembled?

There are two main ways of analyzing metagenomic data: direct read mapping and assembly-based. This study uses the former as was explicitly described in this Methods section and the referred SOP. There was also no mention of an assembly being done. Assembly-based analysis would be seriously inappropriate for these kinds of samples as they are high-diversity and low-sequencing-depth, which would lead to very little assembly and poor functional+taxonomic coverage, leading to spurious results – this was also highlighted directly in the Discussion (original lines 363-366).

2) Was there a cut-off to remove reads with shorter base pairs?;

Yes – again, as explicitly described in this Methods section and the referred SOP, Trimmomatic uses a minimum length of 50 bp in the read QC step.

3) What was the total number of reads per sample after trimming and filtering? A table with these numbers would be good in the supplementary material.

This information, with raw numbers and a treatment group key (to aid those downloading from ENA), is now provided as the new Dataset S1.

Statistical analyses: Authors have compared different treatments for taxonomic and functional differences. However, the significance values of these comparisons are not listed in the results. In addition, Student's t-test and ANOVA are not the best tests to use for these comparisons. Tools such as LEfSe or DESeq2 would allow the authors to show which taxa/functional pathways are over-represented or enriched in one treatment over the other. These tools are statistically more robust.

Agreed – we have now included both a LEfSe analysis and a comparison to ALDEx2, as the former is sometimes viewed as too permissive and the latter as too conservative. These have now been applied to all data types (taxonomy+functions+pathways) and form the basis of the new Figs. 2 (bottom row), 3, 4 (center heatmap), 6 and 7.

Minor comment:

Line 391: is "more oft-used MGS" referring to "more often-used MGS"?

Done.

November 1, 2021

Dr. Isabel Miralles
University of Almeria
Department of Agronomy & Center for Intensive Mediterranean Agrosystems and Agri-food Biotechnology (CIAIMBITAL)
Almeria
Spain

Re: mSystems00752-21R1 (Functional and Taxonomic Effects of Organic Amendments on the Restoration of Semiarid Quarry Soils)

Dear Dr. Isabel Miralles:

Thank you for submitting your manuscript to mSystems. We have completed our review and I am pleased to inform you that, in principle, we expect to accept it for publication in mSystems. However, acceptance will not be final until you have adequately addressed the reasonably minor comments raised by the reviewers.

Below you will find instructions from the mSystems editorial office and comments generated during the review.

Preparing Revision Guidelines

Sincerely,

Nick Bouskill

Editor, mSystems

Journals Department
Reviewer comments:

Reviewer #1 (Comments for the Author):

I commend the authors on their revised manuscript. The statistical methods used are clearer, graphs are easier to interpret, and they have included several sections in the discussion that really hone the overall focus of the paper and make interesting contributions to the ongoing discussion surrounding soil metagenomes. Minor comments below:

Line 132 -should this be: the fact "that" gene presence?

Line 153 -should this be: discriminate "among" soil types?

Line 541-542 - Even though the authors have clearly outlined in the introduction that functional gene presence equates to "potential" and not actual function, it should be reinforced here - Metagenome analysis is best for the analysis of microbial community functional "potential". Direct analysis of community function would likely be metatranscriptomics, or actual measurement of the function in question (enzyme activity, etc.). This is just a thought about wording though, as the authors clearly outline the differences in their discussion.

Reviewer #2 (Comments for the Author):

The authors did an excellent job at revising the manuscript and have carefully addressed all reviewers' comments. The authors have now included two new analyses, 1) ALDEx2 and LefSe, and 2) a comparison of top taxa from metagenomic and previously published 16S rRNA amplicon sequencing. These analyses have strengthened the manuscript and the revised manuscript is considerably stronger. In addition, the updated Figures and heatmaps are easy to understand and including the statistical methods used in the Figure, legends make it easy for the reader to know exactly which method was used for the analyses.

There are some minor comments that should be addressed, otherwise well done!

1. The PERMANOVA analyses for PCoA plots (taxonomy, functions, and pathways) have high R² values. However, this has not been emphasized in the results. Including a sentence that different treatments account for 87% of the variation in the taxonomy will further strengthen the significant results.
2. In Fig 4, the darker blue color has a lower frequency compared to the green and yellow. In the first observation, the taxa with blue color appeared to be of higher frequency. Switching the color scale might help with that.
3. I am confused about the center heatmap depicting ALDEx2 and LefSe analyses in Figure 4. Does the blue color match the heatmap legend on the left? If the color is not indicative of frequency, changing the blue color to a color not in the heatmap legend would be better to differentiate between the two analyses.

Response to the authors:

The authors did an excellent job at revising the manuscript and have carefully addressed all reviewers' comments.

The authors have now included two new analyses, 1) ALDEx2 and LefSe, and 2) a comparison of top taxa from metagenomic and previously published 16S rRNA amplicon sequencing. These analyses have strengthened the manuscript and the revised manuscript is considerably stronger. In addition, the updated Figures and heatmaps are easy to understand and including the statistical methods used in the Figure, legends make it easy for the reader to know exactly which method was used for the analyses.

There are some minor comments that should be addressed, otherwise well done!

1. The PERMANOVA analyses for PCoA plots (taxonomy, functions, and pathways) have high R² values. However, this has not been emphasized in the results. Including a sentence that different treatments account for 87% of the variation in the taxonomy will further strengthen the significant results.
2. In Fig 4, the darker blue color has a lower frequency compared to the green and yellow. In the first observation, the taxa with blue color appeared to be of higher frequency. Switching the color scale might help with that.
3. I am confused about the center heatmap depicting ALDEx2 and LefSe analyses in Figure 4. Does the blue color match the heatmap legend on the left? If the color is not indicative of frequency, changing the blue color to a color not in the heatmap legend would be better to differentiate between the two analyses.

Reviewer #1 (Comments for the Author):

I commend the authors on their revised manuscript. The statistical methods used are clearer, graphs are easier to interpret, and they have included several sections in the discussion that really hone the overall focus of the paper and make interesting contributions to the ongoing discussion surrounding soil metagenomes. Minor comments below:

Line 132 -should this be: the fact "that" gene presence?

Done.

Line 153 -should this be: discriminate "among" soil types?

Done.

Line 541-542 - Even though the authors have clearly outlined in the introduction that functional gene presence equates to "potential" and not actual function, it should be reinforced here - Metagenome analysis is best for the analysis of microbial community functional "potential". Direct analysis of community function would likely be metatranscriptomics, or actual measurement of the function in question (enzyme activity, etc.). This is just a thought about wording though, as the authors clearly outline the differences in their discussion.

Done.

Reviewer #2 (Comments for the Author):

The authors did an excellent job at revising the manuscript and have carefully addressed all reviewers' comments.

The authors have now included two new analyses, 1) ALDEx2 and LefSe, and 2) a comparison of top taxa from metagenomic and previously published 16S rRNA amplicon sequencing. These analyses have strengthened the manuscript and the revised manuscript is considerably stronger. In addition, the updated Figures and heatmaps are easy to understand and including the statistical methods used in the Figure, legends make it easy for the reader to know exactly which method was used for the analyses.

There are some minor comments that should be addressed, otherwise well done!

1. The PERMANOVA analyses for PCoA plots (taxonomy, functions, and pathways) have high R² values. However, this has not been emphasized in the results. Including a sentence that different treatments account for 87% of the variation in the taxonomy will further strengthen the significant results.

This has now been added to the taxa section on l.155-156 and the functional section on l.213-214.

2. In Fig 4, the darker blue color has a lower frequency compared to the green and yellow. In the first observation, the taxa with blue color appeared to be of higher frequency. Switching the color scale might help with that.

Not sure what the reviewer means by "the first observation", but the color scale is consistently dark blue for low and yellow for high in both heatmaps on either side – this has been reinforced in the figure legend. The change of the middle section away from blue (see below) might help alleviate the confusion.

3. I am confused about the center heatmap depicting ALDEx2 and LefSe analyses in Figure 4. Does the blue color match the heatmap legend on the left? If the color is not indicative of frequency, changing the blue color to a color not in the heatmap legend would be better to differentiate between the two analyses.

Correct – there was no link to the center ALDEx2+LefSe “heatmap” color-wise, as it was simply one color chosen to represent “on” or “positive”. The center part was also not really a heatmap (since implies a scale of numbers), but simply a binary yes/no indication of significance, and so we have appropriately changed it to black-filled (positive) and empty (negative) dots. This should alleviate any confusion with the other heatmaps as there no longer are any shared colors nor shapes with them and they should be distinct enough to imply they are measuring/indicating something different.

November 5, 2021

Dr. Isabel Miralles
University of Almeria
Department of Agronomy & Center for Intensive Mediterranean Agrosystems and Agri-food Biotechnology (CIAIMBITAL)
Almeria
Spain

Re: mSystems00752-21R2 (Functional and Taxonomic Effects of Organic Amendments on the Restoration of Semiarid Quarry Soils)

Dear Dr. Isabel Miralles:

Your manuscript has been accepted, and I am forwarding it to the ASM Journals Department for publication. For your reference, ASM Journals' address is given below. Before it can be scheduled for publication, your manuscript will be checked by the mSystems senior production editor, Ellie Ghatineh, to make sure that all elements meet the technical requirements for publication. She will contact you if anything needs to be revised before copyediting and production can begin. Otherwise, you will be notified when your proofs are ready to be viewed.

As an open-access publication, mSystems receives no financial support from paid subscriptions and depends on authors' prompt payment of publication fees as soon as their articles are accepted. =

Publication Fees:

We recognize that the video files can become quite large, and so to avoid quality loss ASM suggests sending the video file via <https://www.wetransfer.com/>. When you have a final version of the video and the still ready to share, please send it to Ellie Ghatineh at eghatineh@asmusa.org.

Sincerely,

Nick Bouskill
Editor, mSystems

Journals Department
Supplementary Dataset S2: Accept
Supplementary Dataset S4: Accept
Supplemental FigureS2: Accept
Supplemental FigureS6: Accept
Supplementary Dataset S3: Accept
Supplementary Dataset S1: Accept
Supplemental FigureS1: Accept
Supplemental FigureS5: Accept
Supplemental FigureS3: Accept
Supplemental FigureS4: Accept